# Modeling the oxygen uptake, transport and consumption in an estivating terrestrial snail, *Xeropicta derbentina*, by the Colburn analogy

**Ulf Fischbach**[1], **Heinz-R. Köhler**[2]*, **David Wharam**[3], **Ulrich Gärtner**[1]*

**1** Faculty of Mechanical Engineering, Esslingen University of Applied Sciences, Esslingen, Germany, **2** Animal Physiological Ecology, Institute for Evolution and Ecology, University of Tübingen, Tübingen, Germany, **3** Mesoscopic Physics and Nanostructures, Institute for Applied Physics, University of Tübingen, Tübingen, Germany

* ulrich.gaertner@hs-esslingen.de (UG); heinz-r.koehler@uni-tuebingen.de (HRK)

**Data Availability Statement:** All relevant data are within the manuscript.

**Funding:** UF received a personal PhD grant from the 'Twinning Project' Programme of Tübingen

## Abstract

The present work gives insight into the internal heat management of the respiratory system in the terrestrial snail *Xeropicta derbentina*, which has to cope with extreme climate conditions in its habitat. A realistic model of the lung´s vein system was constructed and the active diffusive surface of capillaries and main vein was calculated and confirmed by geometrical measurements. We here present a model that is able to validate the measured oxygen consumption by the use of the Colburn analogy between mass and momentum transfer. By combining basic diffusion laws with the momentum transfer, i.e. wall shear stress, at the inner wall of the lung capillaries and the main vein, the progression of the oxygen mass fraction in the hemolymph can be visualized.

## Introduction

### Background

Terrestrial pulmonate snails quite often inhabit arid environments, even deserts. In order to tolerate the conditions prevailing in these habitats, they have evolved a number of adaptations [1] among which estivation, i.e. inactivity over a long period of time, is highly effective to withstand dehydration. During estivation the snails usually remain retracted in their shell in a position distant from the hot soil surface, seal the shell opening with partly calcified dried mucus, and reduce their metabolism to a minimum. Some species remain in this state for weeks or even months [2]. Despite the low metabolism, however, survival of the animals depends on a sufficient oxygen supply via the lung, located in the mantle cavity, and a set of associated veins containing the hemolymph. The almost complete isolation of the snail´s soft body from the environment and the reduced metabolic activity pose crucial demands on the performance of the oxygen supply system. A detailed comprehension of the lung function, the limits of oxygen uptake, transport and consumption, and the intrinsic quantitative limits posed by structurally based constraints are thus prerequisites for understanding the limitations of the oxygen supply in land snail species in extreme environments.

University by the State of Baden-Württemberg The funders had no role in study design, data collection and analysis, decision to publish, or preparation of the manuscript.

**Competing interests:** The authors have declared that no competing interests exist.

In our study, we concentrated on the xerophilic hygromiid land pulmonate, *Xeropicta derbentina* (Krynicki, 1836), a species from the Middle East which has been successfully colonizing other arid habitats in recent times. In order to construct a physiological model of oxygen transport through the body of *X. derbentina*, we obtained available information on the relevant parameters, e.g. breathing frequency, breathing volume and oxygen reduction per breath, from the literature and investigated the geometry of the snail´s respiratory system by micro nuclear magnetic resonance (micro-NMR). The resulting integrative physiological model aims at providing ranges for the aforementioned values and helping to further characterize the physiological state of the animal without the need to directly measure the corresponding parameters.

Based on physiological data, the geometry of the lung and the heart with its afferent veins, the oxygen concentrations in artery and vein are calculated as well as the amount of active diffusive surface in the snail's lung. These calculations subsequently lead to a range for the heart rate of *X. derbentina* since the volume flow of hemolymph is directly linked to the absorbed oxygen concentration during respiration.

Here we present a method to calculate the oxygen concentration along the venous system of *X. derbentina* that is based on a realistic model of the venous system derived from the NMR measurements and utilizes the analogy between local shear stress in the venous system and mass transfer between the snail's lung and its hemolymph, known as the Colburn analogy. Finally, the model was validated by comparing the molar diffusion flow through all capillaries and the main vein with the measured oxygen consumption rates of *X. derbentina*. With our model we aim to provide a better understanding of the metabolic processes in a highly adapted organism which is able to tolerate conditions that seem, at first glance, rather unsuitable for survival.

## Oxygen consumption and metabolic scaling of *X. derbentina*

The influence of temperature on oxygen consumption in snails has been investigated by Mason [3], who recorded respiration rates of twelve species at 5˚C, 10˚C, and 15˚C. His findings suggest a general rise in the metabolic rate with increasing temperature with only a single species displaying stagnant oxygen consumption at the highest test temperature. This tendency has also been confirmed for two different species of slugs, *Limax maximus* and *Philomycus carolinianus* [4]. Schmidt-Nielsen and colleagues [5] investigated the influence of temperature on *Sphincterochila boissieri*, a snail inhabiting the deserts of the Middle East and thus facing a very dry and hot climate. The authors showed that higher temperatures resulted in higher oxygen consumption and that this relationship mirrored the data recorded for non-desert snails. They concluded that no metabolic adaptation to temperatures up to 35˚C was present, but, as the lethal temperature for *S. boissieri* is known to be 50–55˚C, higher experimental temperatures up to this limit should be investigated. Furthermore, they noticed remarkable variations in consumption ('oxygen bursts') which appeared periodically and were believed to be intrinsic since they occurred under constant external conditions and may, therefore, be characteristic for the metabolism of *S. boissieri*. Herreid [6] has also observed this phenomenon in *Otala lactea* from Morocco. His results showed highly variable oxygen consumption rates of six individuals after arousal from dormancy which was triggered by increased humidity. Riddle [7] compared oxygen consumption between the desert snail *Rabdotus schiedeanus* and the garden snail *Helix aspersa* and found a general rise in oxygen consumption for temperatures up to 25˚C. Beyond 25˚C a metabolic regulation became visible for both species, and oxygen consumption declined, contrary to the results of Schmidt-Nielsen et al. [5]. Riddle concluded that the 'depressed metabolism (. . .) at high temperatures is adaptive in conserving metabolizable

energy'. This was especially evident for the desert snail *R. schiedeanus* that, in general, showed a lower oxygen demand. These studies were confirmed by Dallas and colleagues [8] showing that individuals of *Trigonephrus sp*. had a significantly higher oxygen demand at 15˚C than at 25˚C, corresponding to an 'active' and an 'inactive' state of these Namibian desert snails. In comparison with the similarly sized species *Helix aspersa* and *Otala lactea*, oxygen consumption was significantly lower showing the metabolic adaptation of *Trigonephrus* to much warmer climatic conditions. Riddle [9] demonstrated that atmospheric humidity has a strong influence on the oxygen consumption of the desert snail *R. schiedeanus*, resulting in significantly higher consumption for increasing relative humidity. The comparison with *Helix aspersa* additionally revealed the lower metabolic rates for the desert snail at high temperatures when humidity was very low [7]. Comparative oxygen measurements for banded and unbanded morphs of *Cepaea hortensis* revealed the oxygen consumption to generally increase between 5–15˚C and to stagnate for the unbanded morph at 25˚C [10].

Nopp [11] reported the difference in oxygen consumption between starving, dry-sleeping, and 'active' pulmonates. Starving *Arianta arbustorum* individuals showed significantly lower oxygen consumption rates than fresh-fed conspecifics. Furthermore, dry-sleeping individuals further reduced their oxygen consumption to approximately half of the 'hungry-level' after 2–4 days of inactivity. Blazka [12] and Nopp [11] also reported that oxygen consumption in pulmonate land snails drops drastically at the beginning of estivation and subsequently remains almost constant from the second week of estivation onwards.

It is well known that the weight of an organism represents a very important factor in determining the metabolic rate. At the end of the 19th century Rubner [13] noticed that the weight-specific metabolic rate of homoeothermic animals decreases with increasing body size. If, however, metabolic rate is calculated per unit body surface almost constant values can be obtained. This is known as the 'surface rule' stating that the surface of two geometrically similar bodies can be expressed by the 2/3 power of weight multiplied by a constant.

A general mathematical expression for the dependency between metabolic rate and weight, the metabolic scaling, is given by the well-known allometry formula:

$$MR(m) = b \cdot m^{\alpha} \tag{1}$$

where *MR* is the metabolic rate, expressed by oxygen consumption, *m* is the body weight, α is the scaling exponent, and *b* is a specific constant. Eq 1 can also be written as:

$$\log MR = \log b + \alpha \cdot \log m \tag{2}$$

This results in a linear relationship between log*MR* and log*m* with the slope α. If α = 1 Eq 2 represents an isometric relationship, which results in a doubling of the metabolic rate as the weight doubles. If α = 2/3 one directly obtains the relationship corresponding to the surface rule. Kleiber [14] reviewed applications of the surface law in zoophysiology and showed that, for a certain group of mammals, the metabolic rate was proportional to the power of 3/4th of the body weight. According to that study, the surface rule cannot be the sole explanation for the metabolic activity of animals but the behavior of a species has to be considered as well. Another general review of metabolism and body size is given by von Bertalanffy [15]. The author supposed that three different 'metabolic types' exist and that each species essentially belongs one of these types. In the first type metabolism and growth is described in such a way that the metabolic rate is proportional to the surface or the 2/3 power of weight following the assumptions of the surface rule. In the second type the rate is proportional to weight itself and, in the third type, an intermediate proportionality between surface and weight is found with 1>α>2/3. In recent years the generality of the 2/3 or the 3/4 power of weight metabolic

scaling has been questioned [16, 17]. Glazier [18] reviewed published data and showed that significant deviation exists from the 2/3 or 3/4 scaling model among mammals (0.38–1.11), squamate reptiles (0.27–1.26) and invertebrates (-1.2–2.05) (α became negative when juveniles were included). These deviations could not be explained by theoretical models, such as the resource-transport-network model [19, 20], which predicts an exponent α of 0.75, from molecules to whole organisms, and describes the way 'materials are transported through space-filling fractal networks of branching tubes'. An alternative model based on cell size describes the change in metabolic scaling as a result of the way body size changes. Growth driven by an increasing cell number results in an isometric scaling of metabolism, whereas growth via an increasing cell size results in an exponent α = 0.67 [16]. The results of Chown and colleagues [17] revealed a broad variation of α (0.67–1) for an interspecific analysis of eight ant species, which also corresponded to the cell growth types of Kozłowski et al. [16]. Furthermore, ontogenetic factors were investigated, such as phases of fast growth and reproduction that increase metabolic scaling, resulting in the finding that the exponent may change during the course of ontogeny [21]. A decade ago, Glazier [22] reviewed data of 19 ectothermic species and showed that the mean value of the metabolic exponent_ was significantly higher for 'active' animals (0.918) compared to 'inactive' ones (0.768).

For poikilothermic invertebrates, contradictory results concerning the exponent have been published. Liebsch [23] investigated three species of *Helicidae* showing a direct proportionality to weight and hence α = 1, and the measurements carried out with *Cepaea vindobonensis* by von Bertalanffy and Müller [24] support this relationship. This was confirmed by the findings of Kienle and Ludwig [25] who measured oxygen consumption in *Helicella candicans*. In contrast to these results Wesemeier [26] found the average slope of the curve in a log-log plot for four different species of land snails to be α = 0.76. Wesemeier also criticized the results of Liebsch [23] for not taking the state of locomotion into account. By comparing *Helix pomatia* at different levels of activity he showed that phases of resting and movement have an influence on the slope α resulting in values of 0.8 for moving snails and 0.71 for resting individuals. Furthermore, he noted that Liebsch [23] had calculated the regression coefficient as an average value for different species without accounting for interspecific variation. The results of Wesemeier [26] for α in pulmonated land snails varied between 0.71 and 0.85 showing that there might not be an exact correlation between weight and metabolism, such as stated by the surface law, but rather a species-specific relationship. In addition, the author confirmed the result of Kienle [27] where the lung surface of *Helix pomatia* was found to be proportional to the 0.74th power of weight. However, it is still unclear whether this is an intrinsic and therefore species-specific genetically fixed feature or if metabolism itself is limiting the growth of the lung surface. Steigen [10] presented results of *Cepaea hortensis* for energy metabolism at temperatures from 5–25°C showing a continuous decrease of α for increasing temperatures ranging from 0.96 to 1.60. Mason [3] found a linear relationship on a double log plot between weight and oxygen consumption in an interspecific comparison between twenty terrestrial snail species. However, this trend was only found for species with large adult size. The author presented regression coefficients of 0.74, 0.65, and 0.71 for their slope at 5°C, 10°C, and 15°C, respectively [3]. Czarnołeski and colleagues [28] showed that phases of slow and fast growth in ontogeny affect the relationship between metabolism and size in *H. aspersa* resulting in an almost isometric relation (α = 1) for the initial fast-growing ontogenetic phase and a lower value of α for the slow-growing phase. Similar variations in α due to different developmental stages of 421 individuals of *Cornu aspersum* were documented by Gaitán-Espitia et al. [29]. Air-breathing water snails (*Basommatophora*) have been investigated by Berg and Ockelmann [30] who found variations in α ranging from 0.45 to 1.0. They concluded that seasonal changes were the reason for this and also pointed out that 'the relation, oxygen consumption to body

size, is not a fixed, unchangeable quantity characteristic for all species (. . .)'. Additionally, Duerr [31] presented measurements for *Lymnaea stagnalis*, another airbreathing water snail, and found a direct proportionality of respiration to its weight. Acclimation to different temperatures within an appropriate time span is known to pose a physiological problem for some species that may result in different oxygen consumption rates, as reported by a number of authors [4–6, 10, 32]. This effect, however, is not addressed in the present study. Clearly, the discussion of the interrelationship of temperature, body size, and metabolism in terrestrial snails has a long-standing tradition. In this context, the present investigation aims at elucidating and modelling the influence of temperature on the oxygen consumption of *X. derbentina*.

## Material and methods

### Oxygen consumption measurement

The present investigation focuses on *Xeropicta derbentina* (Krynicki 1836), a hygromiid land snail. Specimens were collected from an untreated meadow on private property close to Modène, department Vaucluse, southern France ($N44˚6.055'E5˚7.937'$) between August and early November 2013. The owner neither applied any pesticides to the landsite nor used it as farm land. All snails collected were transported to the laboratory, kept hydrated and fed lettuce and baby porridge. To assure constant starting conditions for all measurements snails were preconditioned in a separate climate chamber that reproduced the 24h humidity and temperature cycle characteristic for an average day in August in southern France. Immediately before the measurements, the individuals' masses and shell diameters were measured. In this study only uniformly white-shelled individuals—by far the most abundant morph—were used in accordance with the shell coloration pattern classification used by Köhler and colleagues [33]. Measurements were carried out for 20˚C, 30˚C, and 38˚C for three different size-groups of snails, group 1 with 0.65–0.85 cm shell diameter (n = 23), group 2 with 0.9–1.0 cm (n = 22) and group 3 with 1.0–1.25 cm (n = 25). For each size group and for each temperature at least n = 7 snails were analyzed to assure a minimum of required statistical reliability.

The necessity to both observe the snails and measure oxygen consumption in parallel as well as the need for a high accuracy required a novel-construction of a purpose-built micro-respirometer. For the measurement of oxygen consumption, a cylindrical aluminum chamber containing a defined air volume was constructed. To capture both 'inactive' and 'active' phases independently, snails were measured individually. The top of the chamber was equipped with an oxygen sensor O2 tracer which is based on an electrochemical cell with a sensitivity of 0.01% O2. The principle of this sensor is similar to a battery or a fuel cell, where chemical energy is transformed into a continuous electric current. A varying concentration of the reactant oxygen results in a variable corresponding current. For a constant total pressure, which was essential for the measurements, a defined rate of oxygen, according to the partial pressure of oxygen $p_{O2}$, diffuses through a membrane and participates in the reaction. The top of the chamber was sealed with acrylic glass that allows the observation of the snail's activity. A custom plastic cover was designed, in which a small webcam and an LED were integrated for the recording of the animal's activity. This permitted the separate determination of oxygen consumption rates both in 'active' and 'inactive' phases. The entire respirometer chamber was then placed in a water bath to adjust the environment for the different test temperatures. To minimize the thermal impact of the observation webcam only 10 pictures per hour with a flashing LED were taken, otherwise LED and webcam were turned off. Furthermore, gas pressure, temperature and humidity inside the chamber were monitored. On the basis of the measured relative humidity the dry gas pressure was calculated by subtracting the water vapor pressure from the total pressure. The water vapor pressure is defined by the following product

of relative humidity $\Phi$ and saturated vapor pressure $e(T)$:

$$p_{water\ vapour} = \phi \cdot e(T) \tag{3}$$

The saturated vapor pressure and its dependency upon temperature is described by the Clausius-Clapeyron equation:

$$\frac{\partial e(T)}{\partial T} = \frac{\Delta H_{vap} \cdot e(T)}{R \cdot T^2} \tag{4}$$

with $\Delta H_{vap}$ as the molar enthalpy of vaporization and $R$ is the universal gas constant. Due to the dependency of $\Delta H_{vap}$ upon temperature an approximation formula is used to determine $e(T)$, the so-called *Magnus formula* [34]:

$$e(T)[Pa] = 6.112 \cdot e^{\frac{17.62}{243.12+T}} \cdot 100 \tag{5}$$

with the temperature $T$ given in degree Celsius. Total dry gas pressure, temperature and air volume finally yield the total molar amounts of the substances $n_{total}$ inside the chamber, as stated by the ideal gas equation:

$$n_{total} = \frac{p_{total,dry} \cdot V_{chamber}}{R \cdot T_{chamber}} \tag{6}$$

Using *Dalton's law* and the definition of the spatial fraction of oxygen $r_{O_2}$ in the mixture, one obtains for the amount of oxygen:

$$n_{O_2} = \frac{p_{O_2} \cdot V_{chamber}}{R \cdot T_{chamber}} = \frac{r_{O_2} \cdot p_{total,dry} \cdot V_{chamber}}{R \cdot T_{chamber}} \tag{7}$$

The time evolution of each measurement revealed the variation of the amount of oxygen in the units of $[mol \cdot s^{-1}]$, which is equal to the oxygen consumption of each specimen. Relative humidity was not controlled during the experiments but was always in the range of 25–50% depending on ambient conditions. The validity of the measurements was confirmed by a base-line record of a constant level of oxygen for the empty chamber at each temperature. The oxygen sensor, which was directly screwed into the acrylic glass at the chamber's top, was provided with a hardware implementation for temperature correction, integrated in its outer casing, and thus located outside the measurement chamber. To minimize environmental temperature gradients in this hardware that would immediately influence the oxygen signal, the measurement took place in an insulated cabinet. The measurement chamber was then placed in an additional plastic case that was used as a water bath and the water circulation was realized via an external setup. This arrangement resulted in a very stable microclimate inside the cabinet since the measurement chamber was conditioned by the temperature of the circulating water and was properly insulated from external influences (Fig 1).

Additionally, baseline measurements were taken with an empty chamber and a linear correlation ($r = -0.856$, Pearson) was obtained between the oxygen signal and the temperature that was measured at the position of the outer casing of the oxygen sensor (compare to Fig 1). This was subsequently used to correct the measured oxygen signal for unavoidable temperature drifts. With this correction the oxygen signals for baseline measurements were shown to vary in a range of 0.01% $O_2$ level, representing the maximum accuracy of the oxygen sensor and demonstrating the high accuracy of the measurements. The gas pressure sensor was placed directly above the water level to reduce the hose length outside of the water bath, and thus minimize the influence of slight temperature gradients. In order to avoid the impact of

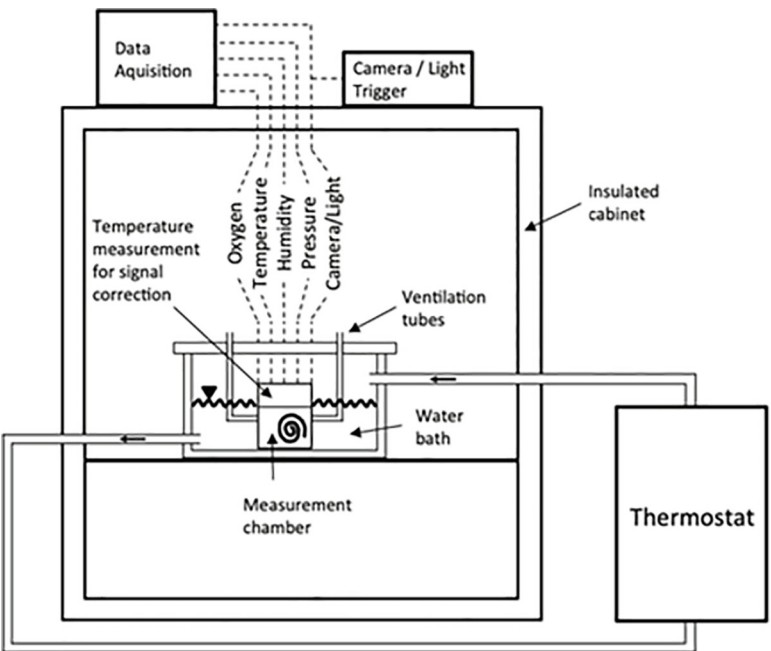

**Fig 1. Schematic depiction of the camera-equipped micro respirometry system.**

different movement activity on the measurements, only data recorded for individuals in their 'inactive' phases were used for this study.

## Statistics

Data were checked for normal distribution using the Shapiro-Wilk W-Test in JMP 11.0 (SAS Institute Inc.) after $4^{th}$ root transformation. Homogeneity of variances was guaranteed by non-significance in Levene's tests. Significance was then checked by ANOVA with a post-hoc Tukey-Kramer test.

If not specified otherwise p≤0.05 was defined significant (*), p≤0.01 was highly significant (**), and p≤0.001 was conclusively (***) (very highly) significant.

On the basis of fitted power series a two-parameter function has been determined with Excel 2010 (Microsoft Inc., Redmond, WA, USA) representing oxygen consumption depending on shell free weight and temperature to unify both dependencies.

## Results and discussion

### Influence of weight on the metabolism at different temperatures

This potential interrelationship was investigated using the shell-free weight of each animal. Therefore, empty shells of *X. derbentina* were weighed and correlated to their outer shell diameter yielding an approximately cubic fit (n = 26, $R^2$ = 0.981) that was used to subtract the shell weight from each specimen. The analysis of intraspecific weight influence on oxygen consumption required appropriate fitting of the data points, which was achieved with the software package MATLAB R2013b (Mathworks, Natick, MA, USA) according to the power function $MR = b \cdot m^{\alpha}$.

For 25°C the fitting procedure resulted in $\alpha$ = 1.221 and $b$ = 2.835·$10^{-13}$ with $R^2$ = 0.875 (Fig 2). The uncertainty in coefficients is given by the 95% confidence intervals of 1.000–1.442 and 9.856·$10^{-14}$–8.157·$10^{-13}$ for $\alpha$ and $b$, respectively.

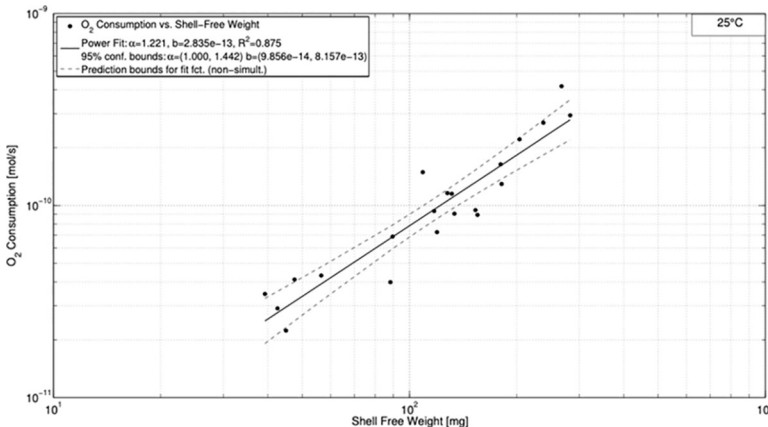

**Fig 2. Oxygen consumption *vs.* shell free weight for 25˚C.**

The fit for 30˚C resulted in $R^2 = 0.566$ for $\alpha = 1.033$ and $b = 8.216 \cdot 10^{-13}$ (Fig 3). The uncertainties for $\alpha$ and $b$ were higher compared to the measurements at 25˚C given by the intervals of 0.633–1.432 and $1.203 \cdot 10^{-13}$–$5.611 \cdot 10^{-12}$ for $\alpha$ and $b$, respectively. The power fit function for 38˚C revealed its maximum of $R^2 = 0.362$ for $\alpha = 0.689$ and $b = 2.18 \cdot 10^{-12}$ (Fig 4). As before, uncertainty increases in comparison to the data at 30˚C (0.294–1.083 and $3.291 \cdot 10^{-13}$–$1.444 \cdot 10^{-11}$ for $\alpha$ and $b$, respectively).

In Table 1 we compare the oxygen consumption of *X. derbentina* with literature data for related land snail species. Only studies were chosen which used the shell-free weight of the corresponding snail species to calculate weight-specific consumptions. Shell-free weight specific values of the present work have been calculated for the mean oxygen consumption value of each size group at a test temperature of 25˚C. The comparison shows that the measured values of oxygen consumption on *X. derbentina* agree well with the results of other snail species, being of the same order of magnitude.

## Weight/metabolism relationship

The results in the present study demonstrate a clear relationship with temperature, similar to the one observed by Steigen [10], namely a decrease in $\alpha$ for increasing temperatures. For

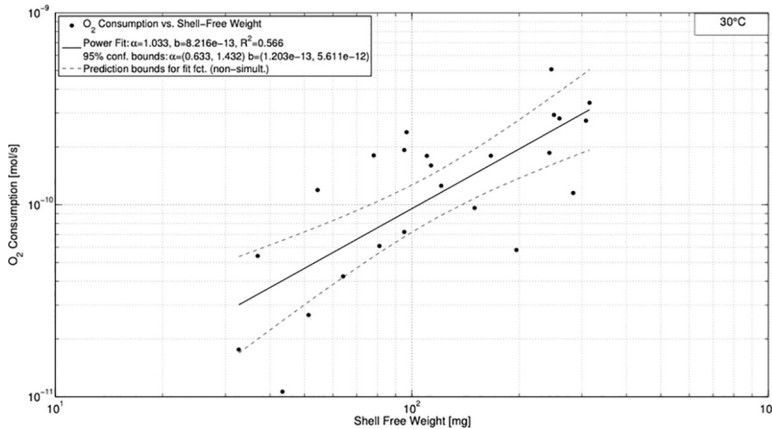

**Fig 3. Oxygen consumption *vs.* shell free weight for 30˚C.**

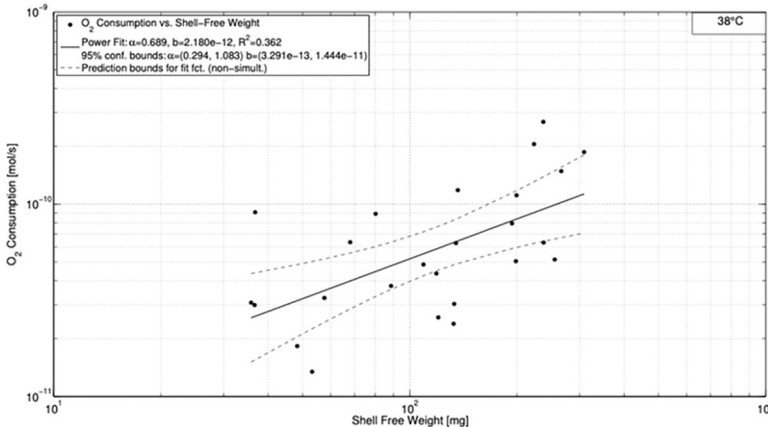

**Fig 4. Oxygen consumption *vs*. shell free weight for 38˚C.**

25˚C the value of α was 1.22, which is outside the range postulated by von Bertalanffy [15]. For 30˚C the value of α is approximately 1 indicating a direct proportionality to weight consistent with the results of Liebsch [21] and also Kienle and Ludwig [23]. At the highest temperature of 38˚C, α declines even further to the value of approximately 0.69 that is close to the value for animals obeying the surface rule.

Wesemeier [24] presented results for both 'active' and 'inactive' specimens of *Helix pomatia* and reported an increased value of α for the 'active" individuals that he could not explain. For *X. derbentina* as well as for other helicoid land snail species it is well known that these snails become more and more 'inactive' as temperatures increase. Even though no 'active' snail was measured in the present study, it is conceivable that snails at the lower test temperatures may have a higher state of internal activity than the ones measured at the highest temperature, which are trying to lower their metabolism as much as possible. It is therefore possible that the state of activity, indirectly triggered by temperature, has a decisive influence on α.

It should be mentioned that measurements become less accurate at high temperatures due to unavoidable methodological reasons and, therefore, an evaluation of the accuracy of fit is difficult. The reasons for this are the lower oxygen consumption at higher temperatures and

**Table 1. Overview of oxygen consumption values for different snail species.**

| Species | Condition | Specific oxygen consumption [mol/(mg $\times$ s)] | Reference |
|---|---|---|---|
| *Xeropicta derbentina* | 25˚C, dormant, group 1 | $7.013 \times 10^{-13}$ | present study |
| | 25˚C, dormant, group 2 | $8.297 \times 10^{-13}$ | |
| | 25˚C, dormant, group 3 | $1.008 \times 10^{-12}$ | |
| *Bulimulus dealbatus* | 22˚C, dormant | $2.271 \times 10^{-13}$ | Horne (1973): Am. J. Physiol. 224:781–787. |
| *Radbotus schiedanus* | 25˚C, dormant | $3.179 \times 10^{-13}$ | Riddle (1975): Comp. Biochem. Physiol. A. 51: 579–583. |
| *Helix aspersa* | 25˚C, dormant, 60% relative humidity | $4.996 \times 10^{-13}$ | Riddle (1977): Comp. Biochem. Physiol. A. 56: 369–373. |
| *Radbotus schiedanus* | 25˚C, dormant, 60% relative humidity | $2.044 \times 10^{-13}$ | |
| *Otala lactea* | 20˚C, dormant | $2.767 \times 10^{-13}$ * | Herreid (1977): J. Exp. Biol. 55: 385–398. |
| *Otala lactea* | 23–25˚C, dormant, 32% relative humidity | $1.555 \times 10^{-13}$ | Barnhart & McMahon (1987): J. Exp. Biol. 128: 123–138. |

* specific consumption was calculated based on the provided average shell weight.

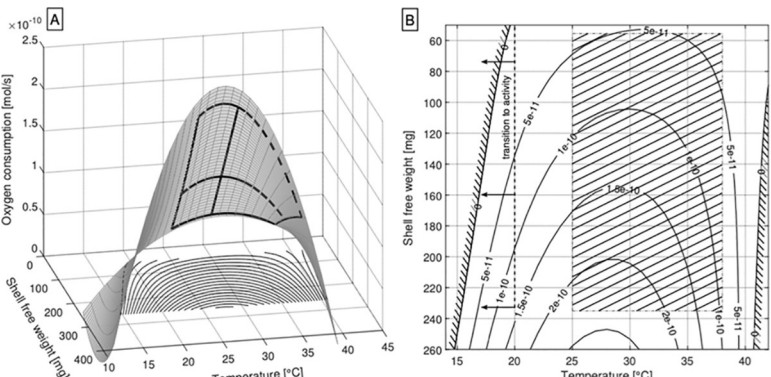

**Fig 5.** A: Response surface calculated for oxygen consumption vs. temperature and shell free weight. B: Two-dimensional plot of constant $O_2$ consumptions (isoboles) in a temperature vs. shell free weight diagram. Zero isoboles represent the theoretical limits (zero $O_2$ consumption) deduced by extrapolation. The hatched area shows the region covered by the measurements.

the increasing sensitivity of the respirometric system at higher temperature differences between measurement chamber and the outer environment. Another reason may be given by increased thermal stress on *X. derbentina* that arises at higher temperatures. Under the assumption that this effect starts primarily in only a few individuals and not in the entire group of animals at the same time, the overall variance increases and therefore $R^2$ declines.

A complete view of the dependence of oxygen consumption upon temperature and body size is given in Fig 5A. The resulting equations of the power fits from Figs 2–4 have been used to calculate oxygen consumptions for each temperature using the mean shell free weights of each shell size category. For the representation of oxygen consumption of as a function of shell free weight and temperature the allometric law has been modified. The surface for oxygen consumption can then be represented by a function of the form _

$$
\begin{aligned}
\dot{O}_2(m, T) = {} & a_1 + a_2 \cdot m + a_3 \cdot m^2 \\
& + (b_1 + b_2 \cdot m + b_3 \cdot m^2) \cdot T \\
& + (c_1 + c_2 \cdot m + c_3 \cdot m^2) \cdot T^2
\end{aligned}
\tag{8}
$$

with coefficients according to Table 2.

The resulting plot reveals conditions under which the oxygen consumptions theoretically would yield negative values, representing theoretical limits of physiology. Additionally, isobolic curves of constant oxygen consumptions are depicted in Fig 5B displaying the "regions" of predicted physiological limits (zero or negative oxygen consumptions for both very low and very high ambient temperatures). Even though these "limits" derive from extrapolations and are, thus, purely theoretical they fit rather well to behavioral and biochemical adaptations of *X. derbentina* to environmental temperatures. The lower temperature "limit" (15–20°C) corresponds to conditions triggering arousal from inactivity resulting in the transition into active

**Table 2. Coefficients of the calculated response surface.**

|       | 1 | 2 | 3 |
|-------|-----------------------------|--------------------------|------------------------|
| $a_i$ | -1.1549 x $10^{-11}$ | -5.9971 x $10^{-6}$ | 9.2619 x $10^{-3}$ |
| $b_i$ | -6.5462 x $10^{-13}$ | 4.6534 x $10^{-7}$ | -4.9078 x $10^{-4}$ |
| $c_i$ | 3.2832 x $10^{-14}$ | -7.7807x $10^{-9}$ | 6.2120 x $10^{-6}$ |

movement and feeding activity. However, this region is associated with some degree of uncertainty due to its rather distant extrapolation from measured data as indicated by the hatched surface. The upper temperature "limit" (slightly above 40˚C) corresponds astonishingly well to the temperature that induces the maximum heat shock protein (Hsp70) response after 8h exposure [35]. For even higher temperatures, the Hsp70 response in *X. derbentina* starts to be overwhelmed, indicating the biochemical limits of this species, as also extrapolated for the O2 consumption in the presented model.

## Diffusive and convective transport of oxygen through the venous system

The transport of oxygen in the snail's hemolymph, relies upon two mechanisms: oxygen is either physically dissolved in the aqueous plasma or chemically bound to hemocyanin, the respiratory protein of molluscs.

$O_2$ inhaled into the lung passes the state of physical solubility after it has diffused through the venous membrane and comes into contact with the transportation fluid (hemolymph). Only in this solute phase the is oxygen able to reach its reaction partner hemocyanin. After the oxygen has been transported to the tissues, the molecule dissociates from this bond again and reenters the phase of physical solubility to perform the exchange between hemolymph and tissue [36].

The absorption in the chemical bond takes place with the transport in the solute phase making the hemocyanin act like a buffer. When this buffer is saturated additional oxygen can pass from the gaseous into the dissolved phase until both systems are saturated. However, the amount of oxygen that is transported by hemocyanin is much higher than the amount of dissolved $O_2$.

In the following, the physical principles and basic calculation methods are presented that we have used to predict values for the concentration of oxygen in the vein and artery of *X. derbentina* as well as the heart rate (= heart beat frequency) and the average thickness of the venous membrane through which the diffusion takes place. The results of this modelling approach define boundary conditions for the diffusive transport of oxygen through the venous system as well as the distribution of oxygen concentration along the main vein and capillaries. The latter is achieved by an application of the Colburn analogy that is used to calculate oxygen concentrations based on local wall shear stress.

## Henry's law

The solubility behavior of volatile substances is described by *Henry's law*. It states that an equilibrium exists between the concentration of a solute gas and the partial pressure of the gas above the liquid surface while volatiles are constantly exchanged between gaseous (subscript "gas") and the aqueous (subscript "aq") phase. For a constant temperature the solubility of a gas is proportional to its partial pressure with the constant of proportionality $k_{H,cp}$ [*mol· l⁻¹· atm⁻¹*] being the Henry constant specified for each liquid-gas combination:

$$c_{aq} = k_{H,cp} \cdot p_{gas} \tag{9}$$

The Henry constant depends on the temperature resulting in a generally decreasing solubility for increasing temperatures. Henry's law can be written differently corresponding to the conversion that is needed and, therefore, different constants for the equation exist. The subscript "*cp*" in Eq 9 indicates the connection between the value of the concentration *c* in the solution and the value of the partial pressure p of the gas. *Henry's law* can also be written in order to link the value of concentration on both sides. In this case the constant holds the index "*cc*" indicating that it represents a value for the conversion between two concentrations. A

comprehensive list of *Henry's law* constants and the corresponding conversion factors can be found in the work of Sander and colleagues [37, 38].

## Oxygen concentration in vein and artery

Oxygen partial pressures for vein and artery have been investigated for the pulmonate land snail *Helix pomatia* by Mikkelsen and Weber [39]. Unfortunately, for *X. derbentina* no measurements have been reported yet. However, as the respiration apparatus of both species is comparable the values obtained for *H. pomatia* may well be similar to the oxygen pressures in *X. derbentina*'s vein and artery. The following calculations are based upon this assumption. Additionally, it is assumed that the physical properties of the snail's hemoolymph are very similar to its main component, water, and, therefore, values for the Henry constants $k_{H,cp}$ and $k_{H,cc}$ are used for the solubility of oxygen in water found in Sander [35].

Mikkelsen and Weber [39] report values of $p_{O_2}$ = 2399.8 Pa and $p_{O_2}$ = 12132.3 Pa for *H. pomatia*'s venous and arterial hemolymph, respectively. Using *Henry's law* an oxygen concentration in the vein can be calculated that is based on the physical solubility of oxygen in the snail's hemolymph. Therefore, the constant $k_{H,cp}$ = 1.3 · $10^{-3}$ mol· $l^{-1}$· $atm^{-1}$ is used to convert the partial pressures into molar concentrations. Applying Eq 11 and multiplying with the molar mass of oxygen $M_{O_2}$ = 31.9988 g· $mol^{-1}$ divided by the density of gaseous oxygen at 25°C $\rho_{O_2}$ = 1.308 g· $l^{-1}$ one obtains:

$$C\, O_{2,vein,phys.sol.} = k_{H,cp} \cdot p_{O_2} \cdot M_{O_2}/\rho_{O_2}$$
$$= 1.3 \cdot 10^{-3}\, mol \cdot l^{-1} \cdot atm^{-1} \cdot 2399.8\, Pa \cdot 31.998\, g \cdot mol^{-1}/1.308\, g \cdot l^{-1} \quad (10)$$
$$= 0.0753\, ml \cdot 100\, ml^{-1}$$

The resulting concentration is interpreted as the volume of dissolved oxygen per 100 ml of hemolymph/water. Analogously, a theoretical concentration in the artery representing the maximum uptake of oxygen by physical solubility alone can be calculated using the oxygen partial pressure given for the artery:

$$C\, O_{2,artery,phys.sol.} = 0.381\, ml \cdot 100\, ml^{-1} \quad (11)$$

The theoretical concentration in the artery resulting from the presence of the complete 20.95% of the ambient oxygen concentration in the snail's lung reveals:

$$C\, O_{2,artery,ambient,phys.sol.} = k_{H,cc} \cdot CO_{2,ambient}$$
$$= 0.666\, ml \cdot 100\, ml^{-1} \quad (12)$$

This result is in good accordance with the value of *0.652 ml· 100 ml⁻¹* for the solubility of ambient $O_2$ (20.95%) in water at 25°C given in Lide [40] and confirms the chosen values of $k_{H,cp}$ and $k_{H,cc}$. The values provided there are given as molar fraction solubilities for a pressure of 1 atm above the solution and, therefore, need to be multiplied with the molar mass fraction $M_{O2}/M_{solvent} = M_{O2}/M_{H2O}$ and the fraction of ambient oxygen concentration 20.95% to obtain the presented value.

Comparing the results of Eqs 11 and 12 shows that the maximal physical solubility cannot be reached with an oxygen pressure of *12,132.3 Pa* resulting in a value of approximately half of the theoretically possible amount. This is most probably due to a reduced oxygen pressure compared to ambient conditions in *X. derbentina*'s lung that effectively lowers the dissolved oxygen in the artery. According to Ortiz-Prado and colleagues [41], who describe this effect for the human lung, the reasons for this decreased oxygen pressure prevailing in the

snail's lung are, mainly, the physiologically caused humidification and the enrichment with $CO_2$.

However, this $O_2$ concentration only results from the physical solubility of the oxygen, and the chemical bonding to hemocyanin has so far been ignored. It is well known that due to this bonding the resulting concentration is substantially raised. If humans were purely dependent on the solubility of oxygen in water our heart rate would be 1,200 min$^{-1}$ requiring a hemolymph flow of 85 l· min$^{-1}$ [42]. Markl [43] states that hemocyanin raises the concentration of oxygen in the hemolymph by a factor of $\Phi$ = 2–4 due to chemical bonding. A final result for the concentration in the artery of *X. derbentina* can be obtained based on the previously calculated physical solubility (Eq 11) and the additional raise by hemocyanin:

$$C O_{2,artery} = \Phi \cdot 0.381 \ ml \cdot 100 \ ml^{-1} \tag{13}$$

This yields a minimum and maximum value for the concentration in the artery $C O_{2,artery,min}$ and $C O_{2,artery,max}$ by evaluating Eq 13 at the minimum and maximum value of the factor:

$$C O_{2,artery,min} = 0.762 \ ml \cdot 100 \ ml^{-1} \tag{14}$$

$$C O_{2,artery,max} = 1.523 \ ml \cdot 100 \ ml^{-1} \tag{15}$$

Assuming that the hemocyanin of the venous blood has completely dissociated its oxygen molecules and the remaining concentration in the vein is completely due to the physical solubility, the previously calculated value of $C O_{2,vein}$ = 0.0753 ml· 100 ml$^{-1}$ may hold true for a final value. Otherwise the minimum and maximum values $C O_{2,vein,min}$ and $C O_{2,vein,max}$ result in:

$$C O_{2,vein,min} = 2 \cdot 0.0753 ml \cdot 100 \ ml^{-1} = 0.151 \ ml \cdot 100 \ ml^{-1} \tag{16}$$

$$C O_{2,vein,max} = 4 \cdot 0.0753 \ ml \cdot 100 \ ml^{-1} = 0.301 \ ml \cdot 100 \ ml^{-1} \tag{17}$$

The amount of oxygen that is absorbed in the snail's lung is the difference between arterial and venous concentrations. Taking the minimum and maximum values of the arterial concentration into account and assuming no complete hemocyanin dissociation of the venous hemolymph the minimum *($\Phi$ = 2)* and maximum *($\Phi$ = 4)* values for the absorbed concentration result in:

$$C_{absorbed,min} = C O_{2,artery,min} - C O_{2,vein,min} = 0.611 \ ml \cdot 100 \ ml^{-1} \tag{18}$$

$$C_{absorbed,max} = C O_{2,artery,max} - C O_{2,vein,max} = 1.222 \ ml \cdot 100 \ ml^{-1} \tag{19}$$

### Heart rate

The heart rate *f* and the resulting transported volume flow of hemolymph are closely connected to the amount of consumed oxygen. The amount of consumed oxygen is exactly the amount that is absorbed by the hemolymph and transported to the organs. Therefore, the amount of absorbed oxygen per volume of hemolymph can also be written as:

$$c_{absorbed} = \frac{\dot{V}_{consumed \ O_2}}{\dot{V}_{hemolymph \ flow}} \tag{20}$$

Based on 3D NMR reconstructions of the anatomy of *X. derbentina* [44] and the subsequent segmentation of specific organs the inner geometrical volume of the heart is known to be $V_{heart} = 8.5mm^3$. Fig 6 shows the reconstructed model of *X. derbentina*.

Furthermore, the hemolymph flow follows from the size of the corresponding ventricle (heart chamber) that fills and empties with each heartbeat and thereby defines the stroke volume. According to Müller [42] the size of a human ventricle is about 35% of the size of the complete heart. Unlike the human heart, which provides a systemic circulation and a pulmonary circulation with two auricles (atriums) and ventricles, most snail species possess a bilocular heart consisting of only a single auricle and a single ventricle that are arranged in a linear sequence [45–47]. Due to this less complex arrangement of the molluscan heart the size of the ventricle is larger than the aforementioned fraction for human beings. To account for the incomplete emptying of the ventricle during a systole DeFur and Mangum [48] used a conservative value of 50% of the heart volume in order to obtain the stroke volume for the genus *Busycon*. Depledge and Phillips [49] used a less conservative value of 70% based on the inner volume of the ventricle.

The uncertainty involved in the segmentation of *X. derbentina*'s heart in the corresponding periphery results in an underestimated value of the heart volume, so that a value of 50% of the heart volume may very well represent the volume of the ventricle including an incomplete emptying of it. This assumption leads to the volume of one ventricle of about $4.25 \cdot 10^{-6}$ l and, therefore, the volume flow of hemolymph results in:

$$\dot{V}_{hemolymph\ flow} = V_{ventricle} \cdot f = 4.25 \cdot 10^{-6} l \cdot f \tag{21}$$

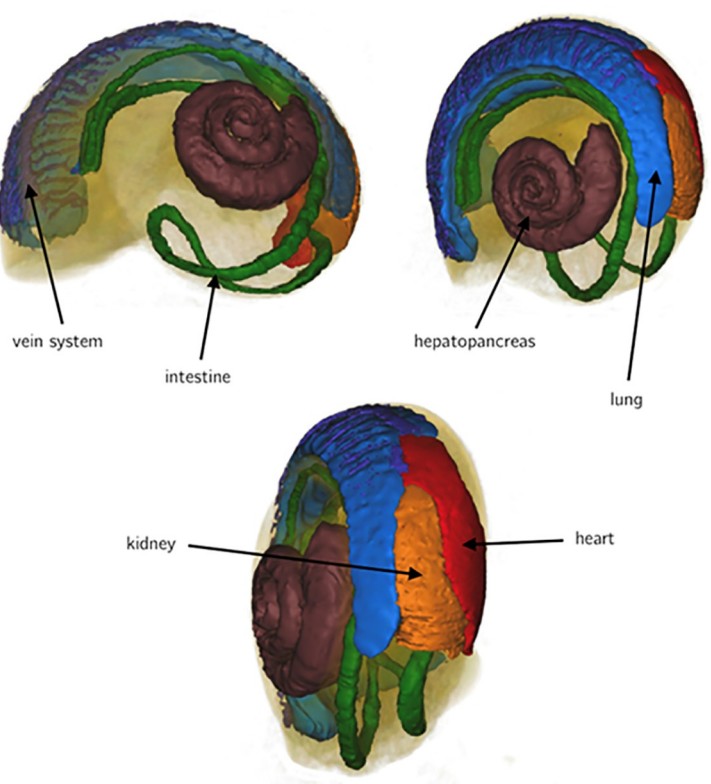

**Fig 6. Reconstructed organs of *X. derbentina* and volume rendering.**

With the calculated range for the absorbed amount of oxygen cabsorbed (Eqs 18 and 19) and the oxygen consumption $\dot{V}_{consumed\ O_2} = 1.395 \cdot 10^{-6}\ l \cdot min^{-6}$ at 25°C the resulting heartbeat frequency range is:

$$f = 2.7 - 5.4\ min^{-1} \tag{22}$$

for 'inactive' specimens of 0.9–0.95 cm shell diameter at 25°C ambient temperature. Schwartz-kopff [50] reported values for the heartbeat frequency of *H. pomatia* in the range of *f = 23.7–30 min⁻¹* measured in an artificial circulatory.However, as these values represent the maximum heart rate corresponding to an 'active' state of the animal the range calculated above is very reasonable for 'inactive' specimens of *X. derbentina*.

## Amount of active diffusive surface

The diffusion flux j is defined as the amount of a substance $\dot{n}$ [$mol \cdot s^{-1}$] passing through the cross section A per unit time t and, thus, has the unit [$mol \cdot m^{-2} \cdot s^{-1}$].

In our approach, it provides a connection between the oxygen consumption of *X. derbentina* and the diffusive surface A, provided by its vein system, over which the diffusion process takes place. The diffusion flux can be calculated by the following equation:

$$j = \frac{\dot{n}}{A} \tag{23}$$

Additionally, *Fick's first law* of diffusion states that the diffusion flux j is proportional to the negative local concentration gradient. For a one-dimensional diffusion process this results into:

$$j = -D \cdot \frac{\partial c}{\partial x} \tag{24}$$

with D [$m^2 \cdot s^{-1}$] being the diffusion coefficient and c [$mol \cdot m^{-3}$] being the molar concentration.

The partitioned vein system extracted from the NMR measurements is rather coarse due to the very small dimension of the capillaries, but it still provides a reasonable model for computer aided simulation. This was realized with the CAD tool CREO Parametric 2.0 (PTC, Needham, MA, U.S.A) by constructing a set of closed surfaces of the vein system that consists of the main vein and the individual capillaries that guide the hemolymph towards the former which subsequently empties into the snail's heart. A mean diameter of all constructed capillaries and of the main vein was calculated and used as a model with constant diameters at all sections. The final model consists of 37 capillaries with a diameter of 0.15 mm and the main vein with a diameter of 0.35 mm (Fig 7). The hemolymph flow in Fig 7 is from the lower left to the upper right of the picture where the heart of *X. derbentina* is located. The structure of two opposing capillaries connected to the main vein in the distal part is changed in the proximity of the heart, due to the asymmetrical structure of the snail´s body.

Expressing *Fick's first law* as a difference equation, setting *Δx = d*, and combining it with Eq 13 permits the calculation of the active diffusive surface *A*. In order to do so the distance *d* over which diffusion takes place has to be known. For the present problem this distance is the thickness of the capillaries and the main vein. These values were measured graphically in Fig 8 using an image from a dissected lung of *X. derbentina* that was achieved by removing the first whorls of the shell from the snail. The dissected snail was then placed above a squared paper with 1 · 1mm unit in order to calculate a scale between measured value and real value (compare Fig 8). The thickness of a capillary wall (membrane) is approximately 0.06mm.

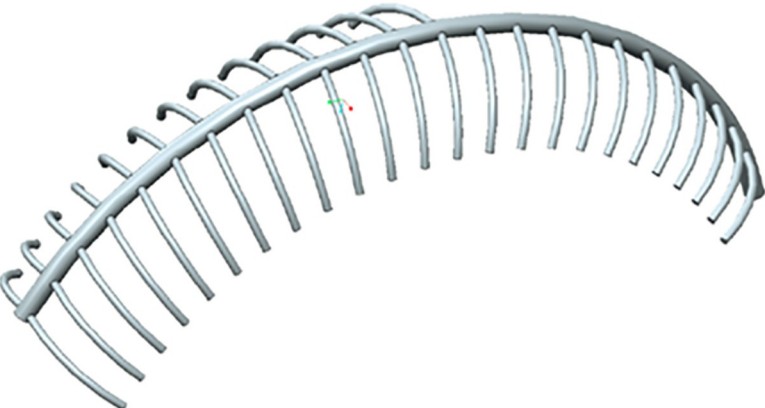

**Fig 7. Geometrical model of *X. derbentina*'s vein system in the area of its lung.** To the left: distal part, to the right: part proximal to the heart.

The epithelial thickness of the main vein was found to be approximately twice as thick with a value of 0.12mm. The values of capillary and main vein diameter used in the model above can also be verified by the graphical measurement, which yields approximately 0.1–0.15mm and 0.3–0.38mm for capillary and main vein diameter, respectively.

*Murray's law* states that the cube of the radius of the parental vessel equals the sum of the cubes of the radii of the daughter vessels [51]. In order to satisfy this law the capillaries would need to have an average radius of 0.11mm, which is in reasonable accordance to the value of 0.15mm. Although the graphically measured values of the diameters differ slightly from the ones chosen in the model, original NMR data showed that the selected diameters very well represent the mean diameters of *X. derbentina*'s vein system, particularly when considering that main vein and capillaries may form irregularly protruded shapes towards the inner lung surface that are not visible from the outside (Fig 8).

Capillaries and main vein are merged with the inner surface of the lung resulting in approximately half-cylindrical shapes of capillaries and main vein (compare Fig 9). By exemplarily choosing a perimeter fraction of 0.5 for capillaries and main vein that actually participates in the diffusive exchange of oxygen between the lung and the hemolymph and using the lengths gathered from the model from Fig 7 a resulting active diffusive surface of 53.9mm$^2$ can be computed from the CAD model in CREO Parametric 2.0.

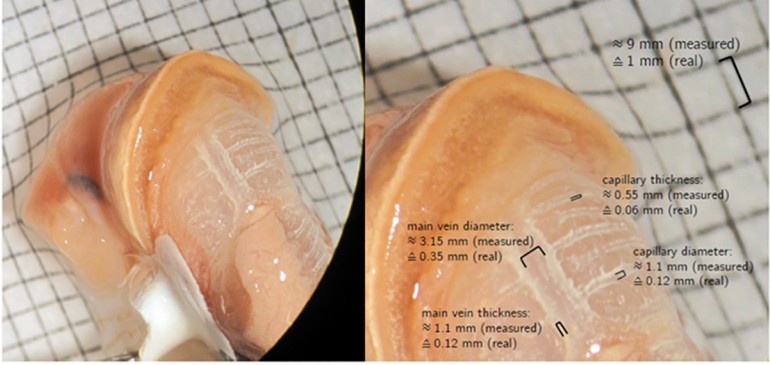

**Fig 8. External view on the dissected lung and measurement of vein diameters.**

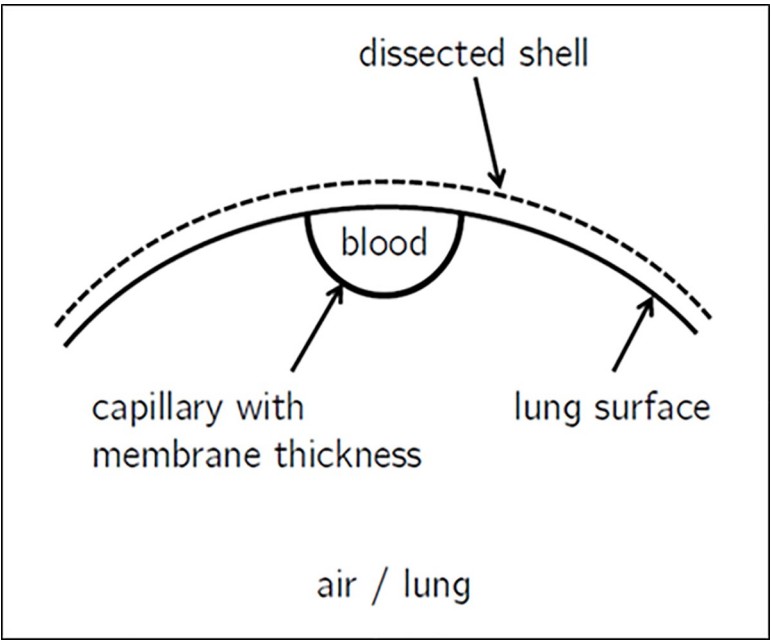

**Fig 9. Geometrical shape of capillaries.**

Considering the reported values of oxygen partial pressure $p_{O_2}$ = 2,399.8 Pa and $p_{O_2}$ = 12,132.3 Pa for venous and arterial hemolymph [39] and the measured oxygen consumption of $1.04 \cdot 10^{-10}$ mol$\cdot$ s$^{-1}$ for medium sized specimens at 25°C a diffusive surface is calculated using Henry's law with the constant $k_{H,cp}$ = 1.3$\cdot$ 10$^{-3}$ mol$\cdot$ l$^{-1\cdot}$ atm$^{-1}$ and the definition of the diffusion flux j from Eqs 23 and 24:

$$A = \frac{\dot{n}}{j} = \frac{\dot{n} \cdot \Delta x}{D \cdot \Delta c} = \frac{\dot{n} \cdot d_{membrane,av}}{D \cdot k_{H,cp} \cdot \Delta p} = 56.2 mm^2 \tag{25}$$

In Eq 25 the following assumptions were made:

- The diffusive length is the mean thickness $d_{membrane,av}$ of capillaries and main vein weighted by their corresponding surface area to account for the presence of both diffusive lengths.

- The venous oxygen partial pressure is the mean value of the ones of artery and vein. The reason for this is the continually increasing $p_{O_2}$ in the snail's vein that merely has a value of $p_{O_2}$ = 2,399.8 Pa at the very beginning of the vein.

- The diffusion coefficient $D$ is the one for diffusion of $O_2$ in pulmonary tissue that is reported by Grote [52] to be $D$ = 2.3$\cdot$ 10$^{-9}$ $m^{2\cdot}$ s$^{-1}$.

The active diffusive surface resulting from geometrical data (53.9mm$^2$) is found to be in good agreement with the one obtained by Fick's law (56.2mm$^2$) that is based on the measured value for oxygen consumption and the oxygen partial pressures reported in the literature as well as on the assumptions stated above.

For the sake of completeness and as an additional comparison the complete dissection of *X. derbentina*'s lung allows for a graphical measurement of its surface on top of a squared paper with 1$\cdot$ 1mm unit. This measurement revealed a total surface size of approximately 50–55 mm$^2$.

Due to the comparatively large thickness of capillary and main vein epithelium the active diffusive surface increases in a way that the resulting size approximates the total surface size of the animal's lung. This result is plausible and can easily be visualized by the unrolling of the half cylindrical vein surfaces in Fig 8. Therefore, the value found by this procedure complements the previous results and confirms the correctness of the geometrical model and the calculation based on Fick's law.

## Analogy of heat, mass and momentum transfer

In the following, the results of the oxygen consumption measurements are compared with a theoretical model. In this model the capillaries and main vein are divided into segments of defined length and the spatial variation of concentration within both systems is calculated by successively passing the values for each segment to the next one. This is first done for all isolated capillaries, which subsequently empty into and mix with the main vein at the corresponding downstream positions resulting in a final concentration that develops along the aorta towards the snail's heart. The necessary input parameters for this procedure only rely on a simple flow simulation through the venous system yielding the velocity and wall shear stress distribution along capillaries and main vein. The presented approach then utilizes an analogy between wall shear stress and mass transfer, which is known as the *Colburn analogy* [53–55]. In contrast to this, a complex CFD simulation that directly takes the diffusive processes of oxygen in water into account requires a much higher computational effort.

This method represents an elegant way to calculate the concentration distribution with only minimal costs of computation time. An evaluation of analytical solutions and correlations based on the Reynolds analogy (upon which the *Colburn analogy* is based) for heat transfer in turbulent tube flow is presented by Webb [56].

The *Colburn analogy* represents a modified and improved version of the well-known *Reynolds analogy*, that is used to predict the heat transfer from momentum transfer. It can be derived by examining the classical problem of convective heat transfer of laminar flow over a flat plate. By non-dimensionalizing the boundary layer conservation equations for momentum and energy it can be shown that for *Prandtl number* Pr = 1 both equations are similar. Due to this similarity a relation between the friction coefficient *cf* and the diffusion mass transfer $j_w$ can be derived [53].

With the definition of the friction coefficient $c_f$,

$$\frac{c_f}{2} = \frac{\tau_w}{\rho \cdot u^2} \tag{26}$$

the *Schmidt number*,

$$Sc = \frac{v}{D} \tag{27}$$

the *Sherwood number*

$$Sh = \frac{\beta \cdot L}{D} = \frac{j_w \cdot L}{\rho \cdot D \cdot \Delta\omega} \tag{28}$$

and the *Reynolds number* respectively

$$Re = \frac{u \cdot L}{v} \tag{29}$$

the friction coefficient $c_f$ can be described as follows:

$$\frac{c_f}{2} = \frac{Sh}{Re} \cdot Sc^{-1/3} \tag{30}$$

The general definition of the diffusion mass transfer $j_w$ [$kg \cdot m^{-2} \cdot s^{-1}$] can then be described as follows:

$$j_w = \rho_w \cdot D \cdot \left. \frac{\partial \omega}{\partial y} \right|_w = D \cdot \frac{1}{L} \cdot \Delta\omega \cdot Re \cdot Sc^{1/3} \tag{31}$$

with L being the characteristic length of the mass transfer.

In respect to the venous system of *X. derbentina*, the characteristic length L of the *Sherwood number Sh* is the length of the corresponding segment of the capillary and main vein, respectively, over which the diffusive mass transfer takes place. Due to the fine segmentation of the venous system the problem approaches the case of flow over a flat plate instead of a tube flow. By this approach the diffusion process in the vicinity of the wall is regarded two dimensional. Therefore, the *Reynolds number Re* is calculated with this characteristic length L (otherwise the diameter of the tube would need to be used) and Eq 31 simplifies to:

$$j_w = \left(\frac{D_1}{\upsilon}\right)^{2/3} \cdot \Delta\omega \cdot \frac{\tau_w}{u_\infty^2} \tag{32}$$

with $\Delta\omega = \omega_w - \omega_2$ as the difference of oxygen mass fraction between the inner membrane wall $\omega_w$ and the snails' hemolymph $\omega_2$. It is important to mention that the diffusion coefficient $D_1$ in Eq 32 is that of oxygen in hemolymph, which will be approximated by that of oxygen in water, which is $D_1 = 2.1 \cdot 10^{-9} \, m^2 \cdot s^{-1}$ [57]. This approximation is justified by the fact that, besides of some cells and dissolved ions, carbohydrates, lipids, glycerol, amino acids, hormones and pigments, hemolymph is basically water. Furthermore, the corresponding area over which diffusion takes place is the inner membrane wall $A_{inner}$.

The diffusion flux transferred into *X. derbentina*'s hemolymph is the same as that which passes through the membrane of the capillaries and main vein. Due to the curved shape of the membrane the area over which diffusion takes place depends on the radius which results in a nonlinear trend of diffusion flow along the radius of the membrane. According to Jischa [50] the equation for the diffusion across a plate is still sufficiently valid if the area given by the mean of the inner and outer radius of the capillary/main vein tube is used for the calculation. The error made by this simplification is less than 3% for the capillaries and less than 5% for the main vein. Therefore, the diffusion flux through the membrane tissue can be written:

$$j = \rho \cdot D_2 \cdot \frac{\Delta\omega_{memb}}{\Delta y_{memb}} = \cdot D_2 \cdot \frac{\omega_{max} - \omega_w}{\Delta y_{memb}} \tag{33}$$

with $D_2$ as the diffusion coefficient for oxygen in pulmonary tissue [52], $\Delta\omega_{memb}$ as the difference of the constant oxygen mass fraction between the lung $\omega_{max}$ and the one at the inner membrane wall $\omega_w$, and $\Delta y_{memb}$ being the membrane thickness. Furthermore, the corresponding area over which diffusion takes place is the area given by the mean of the inner and outer radius of the capillary/main vein tube $A_{mean}$.

Fig 10 illustrates the convective mass transfer model that is used for *X. derbentina*'s vein system. At the bottom of Fig 10 the lung volume is located with its oxygen mass fraction $\omega_{max}$ resulting from ambient conditions.

Adjacent to it is the membrane wall through which oxygen diffuses into the snail's hemolymph. Additionally, the course of mass fraction is schematically depicted. Through the

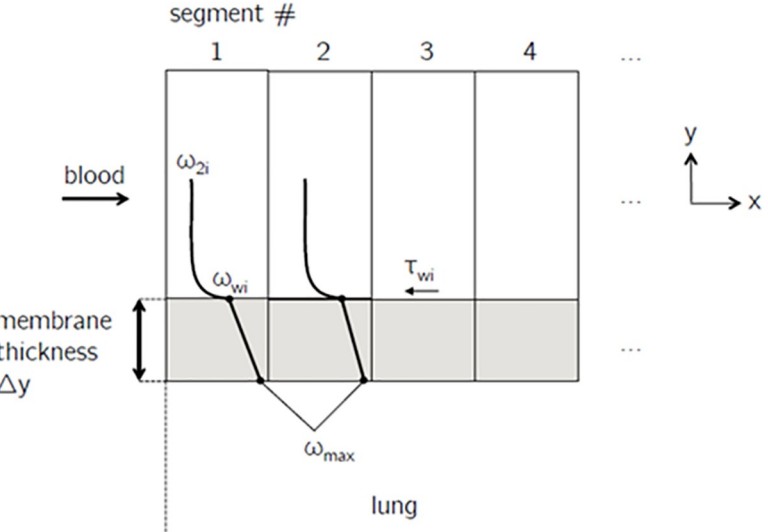

**Fig 10. Convective mass transfer model for *X. derbentina*'s vein system.**

membrane wall a linear decrease of mass fraction is present that transitions into a non-linear course as soon as the convective transport of oxygen into the hemolymph begins. This is evaluated in every segment of each capillary and main vein based on the momentum flux at the wall (shear stress).

Taking the previously mentioned increase of oxygen concentration by a factor of $\Phi = 2\text{--}4$ due to chemical bonding into account it is practical to define a ratio X of concentration that passes from the physical solution into the state of chemical bonding. The following statement can then be made about the physically dissolved mass fraction $\omega 2$ before and after chemical bonding:

$$\omega_{2,after\ chem.bond.} = (1 - X) \cdot \omega_{2,before\ chem.bond.} \tag{34}$$

Therefore, the ratio $X$ is directly connected to the increase of oxygen mass fraction due to chemical bonding $\Phi$ from the relationship:

$$X = \frac{\Phi - 1}{\Phi} \tag{35}$$

It has already been discussed that the ambient oxygen concentration of 20.9% is significantly reduced during the process of inhalation [41].

Thus, the gradient $\omega_{max}-\omega_w$ has to be calculated with a reduced mass fraction $\omega_{max}$ inside the snail's lung. Ortiz-Prado and colleagues [41] report a decrease of up to 20% of the initial ambient oxygen concentration for humans, depending on the altitude. As described before, the authors state that the reasons for even a further reduction in oxygen pressure are mainly the humidification of the breathing air, dead spaces in the lung and the mixing of inspired and expired gases. For *X. derbentina* this decrease may very well be less pronounced than for human beings since the breathing apparatus is far less complex and mitigates these influences. To account for this decrease the oxygen mass fraction $\omega_{max}$ that is present in the snail's lung needs to be multiplied by a factor $\xi$ that, by definition, should lie in the range of 0.6–0.75, rather the latter. Therefore, $\omega_{max}$ is calculated in the following way using the *Henry constant $k_{H,cp}$*, the

molar mass of oxygen $M_{O_2} = 31.99\ g \cdot mol^{-1}$, and the density of water $\rho_{H_2O} = 997\ kg \cdot m^{-3}$:

$$
\begin{aligned}
\omega_{max} &= \xi \cdot p_{O_2,atm} \cdot \frac{M_{O_2}}{\rho_{H_2O}} \\
&= \xi \cdot 1.3 \cdot 10^{-3} mol \cdot l^{-1} \cdot atm^{-1} \cdot 0.2095\ atm \cdot \frac{3.199 \cdot 10^{-2} kg \cdot mol^{-1}}{0.997\ kg \cdot l^{-1}} \\
&= \xi \cdot 8.7387 \cdot 10^{-6}
\end{aligned}
\tag{36}
$$

As an intermediate summary the following variable values have been introduced that need to be adapted to the actual geometrical and physiological setup of *X. derbentina*:

- The surface area fraction of the capillary and main vein tube that is available for diffusion to take place. This surface fraction is an unknown parameter but it should be in the range of 40–60% of the complete inner tube area resulting in approximately half of the cylindrical surface of capillaries and main vein to participate in the diffusion process.

- The ratio X of mass fraction that passes from the physical solution into the state of chemical bonding. This ratio should be chosen in a way that the increase of concentration due to chemical bonding Φ results in a range of 2–4 [43].

- The decrease $\xi$ of the initial ambient oxygen concentration that yields the actual concentration in the snail's lung. As stated above, a reasonable value for this would be 0.75.

Some additional remarks have to be made concerning the Colburn analogy. Numerous textbooks cover the topic of heat transfer in a laminar tube flow with either constant heat flux $q_w$ or constant wall temperature $T_w$ [54, 55, 58–60]. Both cases can easily be transformed to mass transfer by replacing the Nusselt number Nu with the Sherwood number Sh in the same way as it has been done in the present study. These solutions correspond to an integral view of a tube of considerable length. However, the method presented above utilizes the segmentation of the system. Due to its short axial length each segment much more resembles the shape of a flat plate than that of a tube and each mass transfer calculation is only applied for a single segment as well. The method presented calculates the mass transfer for a small segment and adds the sum of all transferred masses into the next segment. Therefore, the tube flow solution is not applicable and the use of the analogy on a flat plate is justified.

The *Colburn analogy*, as well as the *Reynolds analogy*, are only valid for constant wall mass fractions $\omega_w$. At first appearance it seems that this requirement is not fulfilled as the wall mass fraction constantly changes downstream due to the enrichment of oxygen in the hemolymph and its feedback to the membrane wall. Keeping in mind that the use of the *Colburn analogy* only takes place inside a single segment, inside which a constant wall mass fraction is present, the requirement is still fulfilled.

## Determination of oxygen mass fractions

In the following, we describe the basic procedure of calculating the mass fraction of oxygen along all capillaries and the main vein and the corresponding amounts of oxygen that are transferred over each segment.

Additionally, Fig 11 gives an overview of the segmentation of capillaries and main vein that provides the basis for the following calculation procedures. The procedure for each capillary divided into N = 40 segments was:

1. Define $\omega_{2i} = 0$ for the first segment (i = 1)

2. Calculate $\omega_{wi}$ for the first segment

3. Calculate the absolute diffusion mass flow J [kg· s⁻¹] using the mean surface area $A_{mean}$ (between inner and outer radius) and membrane thickness of the corresponding capillary segment by applying Fick's law for mass fractions:

$$J_i = A_{mean} \cdot \rho \cdot D_2 \cdot \frac{\omega_{max} - \omega_w}{\Delta y_{memb,cap}} \tag{37}$$

4. For the following segments (i = 2 . . . 40) use the effective physically dissolved mass fraction after chemical bonding, introduced by the ratio X, for the calculation of $\omega_{2i}$ by relating the absolute diffusion mass flow up to this position to the mass flow of hemolymph at the current position:

$$\omega_{2i} = (1 - X) \cdot \frac{\sum_{n=1}^{i-1} J_n}{\dot{m}_i}$$

$$= (1 - X) \cdot \frac{\sum_{n=1}^{i-1} A \cdot \rho \cdot D_2 \cdot \frac{\omega_{max} - \omega_w}{\Delta y_{memb,cap}}}{\rho \cdot A_{cross-section,cap} \cdot u_{\infty,i,cap}} \tag{38}$$

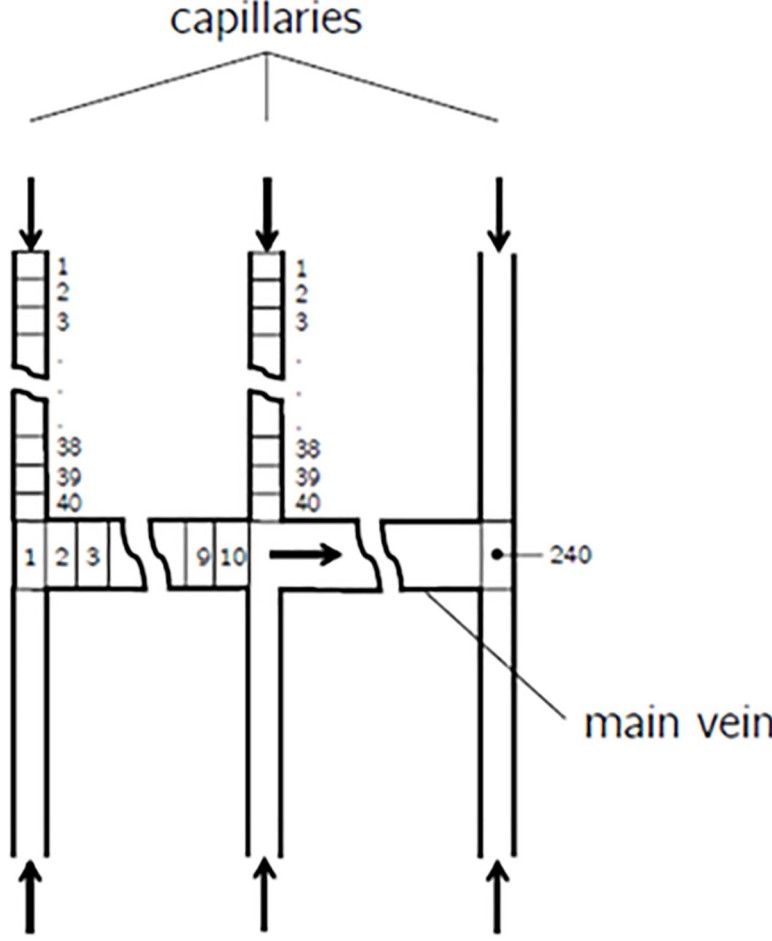

**Fig 11. Segmentation of capillaries and main vein for the vein system.**

5. For the following segments (i = 2 . . . 40) calculate $\omega_{wi}$ with the effective physically dissolved mass fraction after chemical bonding $\omega_{2i}$ obtained from step 4.

6. Calculate the total oxygen mass fraction $\omega_{ti}$ inside the capillary segment:

$$\omega_{ti} = \frac{\sum_{n=1}^{i-1} J_n}{\dot{m}_i} = \frac{\sum_{n=1}^{i-1} A \cdot \rho \cdot D_2 \cdot \frac{\omega_{max} - \omega_w}{\Delta y_{memb,cap}}}{\rho \cdot A_{cross-section,cap} \cdot u_{\infty,i,cap}} \tag{39}$$

7. Finally calculate the sum of all N = 40 diffusion mass flows $J\ [kg \cdot s^{-1}]$ yielding the total mass diffusion flow per capillary.

The procedure for the main vein divided into N = 10 segments between each capillary inflow resulting in a total number of N = 240 segments was:

1. For the first main vein segment (i = 1) calculate the effective physically dissolved mass fraction after chemical bonding $\omega_{2i}$ according to Equation 55 based on the summarized diffusion mass flow from the first two capillaries that have entered the main vein until then related to the mass flow of hemolymph. On the left side of Fig 12 two capillaries enter the main vein at the front, so that the first main vein segment is already mixed with the diffused oxygen flow of these two capillaries.

$$\omega_{2i} = (1 - X) \cdot \frac{\sum_{m=1}^{2} J_m}{\dot{m}_i}$$

$$= (1 - X) \cdot \frac{J_{cap,1} + J_{cap,2}}{\rho \cdot A_{cross-section,main\ vein} \cdot u_{\infty,i,main\ vein}} \tag{40}$$

2. Calculate $\omega_{wi}$ for the first main vein segment (i = 1) using $\omega_{2i}$ from step 1.

3. Calculate the absolute diffusion mass flow for the first main vein segment.

4. (i = 1) using the mean surface area $A_{mean}$ (between inner and outer radius) and membrane thickness of the corresponding main vein segment by applying *Fick's law* for mass fractions:

$$J_i = A_{mean,main\ vein} \cdot \rho \cdot D_2 \cdot \frac{\omega_{max} - \omega_{wi}}{\Delta y_{memb,main\ vein}} \tag{41}$$

5. For the following main vein segments (i = 2 . . . 240) calculate $\omega_{2i}$ based on the summarized diffusion mass flows from all capillaries that have entered the main vein until then related

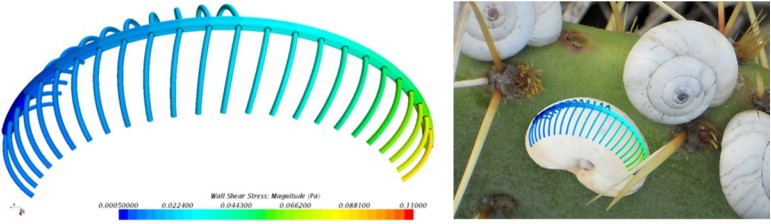

**Fig 12.** Wall shear stress distribution in the vein system, isolated (left) and superimposed on the *in vivo* situation (right).

to the mass flow of hemolymph at the current position, and additionally add the diffusion mass flows from the previous main vein segments from step 3.

$$
\begin{aligned}
\omega_{2i} &= (1 - X) \cdot \frac{\sum J_{cap} + \sum_{n=1}^{i-1} J_n}{\dot{m}_i} \\
&= (1 - X) \cdot \frac{\sum J_{cap} + \sum_{n=1}^{i-1} A \cdot \rho \cdot D_2 \cdot \frac{\omega_{max} - \omega_{wn}}{\Delta y_{memb,main\ vein}}}{\rho \cdot A_{cross-section,main\ vein} \cdot u_{\infty,i,main\ vein}}
\end{aligned}
\tag{42}
$$

6. The value for $u_{1;i;main\ vein}$ in step 1 and step 4 is obtained by the volume-averaged velocities in the capillaries that have entered the main vein until then and the corresponding ratio of the cross-sections between capillary and main vein.

7. Calculate the total oxygen mass fraction $\omega_{ti}$ inside the main vein segment:

$$
\omega_{2i} = \frac{\sum J_{cap} + \sum_{n=1}^{i-1} J_n}{\dot{m}_i} = \frac{\sum J_{cap} + \sum_{n=1}^{i-1} A \cdot \rho \cdot D_2 \cdot \frac{\omega_{max} - \omega_{wn}}{\Delta y_{memb,main\ vein}}}{\rho \cdot A_{cross-section,main\ vein} \cdot u_{\infty,i,main\ vein}}
\tag{43}
$$

8. Finally calculate the sum of all diffusion mass flows $J$ $[kg \cdot s^{-1}]$ transferred in the N = 240 main vein segments and additionally add the sum of all diffusion mass flows $J$ $[kg \cdot s^{-1}]$ from each capillary yielding the total mass diffusion flow.

Additional checks have been implemented for all calculations of the effective physical solute mass fraction after chemical bonding to assure this mass fraction cannot exceed the maximum value $\omega_{max}$ that represents the physical solubility of oxygen in water for the given conditions inside *X. derbentina*'s lung. Therefore, the calculated values are only used if they are smaller than $\omega_{max}$, otherwise they are exchanged for $\omega_{max}$.

## Boundary conditions

The analysis was undertaken for an 'inactive' *T. pisana* individual of 0.9–0.95 cm diameter at 25°C ambient temperature with a corresponding heart rate of 4.8 min$^{-1}$ representing an arbitrary value that covers the upper range derived in this paper. The chosen heart rate in conjunction with the volume flow through the snail's heart given by the derived volume of the ventricle yields a velocity of $u_{main\ vein}$ = 0.0035m$\cdot$ s$^{-1}$ that is used as a boundary condition for the main vein outlet of the model in the CFD simulation. On the inlets of each capillary an ambient pressure boundary condition is applied in order to let velocities in each capillary develop according to the corresponding position along the main vein.

The simulation was performed with the commercial CFD solver Star-CCM+ by CD-Adapco (Melville, NY, U.S.A.). The flow is laminar with a *Reynolds number* of Re ≈ 1.2. Additionally all flow properties of hemocyanin solutions in the simulation are approximated by the corresponding parameters for water.

The distributions and the final sum of all diffusion mass flows yielding the total amount of transferred oxygen depend on the chosen values for the variables defined above. Therefore, reasonable values for these variables were chosen to conduct the calculation presented above.

- The surface area fraction of the capillary and main vein pipe that is available for the diffusion process was chosen to be 60% of the complete inner pipe area (compare Fig 9). This value

seems reasonable because the surfaces of the capillaries and the main vein are likely uneven and larger than an ideal half cylindrical surface. Even the resolution of the 3D-micro-NMR reconstruction only allowed an approximate estimate of the surface of the animal's vein system.

- The ratio X of the mass fraction that passes from the physical solution into the state of chemical bonding is 0.553, resulting in a value for Φ = *2.24*. This follows from the above-mentioned heart rate that depends on the amount of absorbed oxygen and the value of Φ.

- The decrease ξ of the initial ambient oxygen concentration due to inhalation effects was chosen to be ξ = *0.75*. This results in *ωmax = 6.56 x 10$^{-5}$*.

## Wall shear stress distribution

The result of the flow simulation yielding the wall shear stress distribution for the model is given in Fig 12. It is obvious that within each capillary the wall shear stresses are approximately constant neglecting the influence of the run-in distance of the laminar flow. Due to the main vein velocity outlet and the capillaries pressure inlet boundary conditions the volume flow generally increases for capillaries further downstream corresponding to higher values of wall shear stress as it can be observed from Fig 12.

For further analysis the velocity and wall shear stress for each capillary is assumed to be constant. The distribution for the main vein is evaluated at the inlet, at the outlet, and at a central position in order to calculate a quadratic fit that will serve as an approximation to the actual distribution.

The theoretical value for the wall shear stress can easily be derived by the velocity distribution for laminar flow in a pipe (*Hagen-Poiseuille flow*) and the definition of pressure loss for laminar flow. The velocity distribution along the cross-section for a fully developed laminar flow is:

$$u(y) = \frac{\Delta p}{4 \cdot \mu \cdot l} \cdot (R^2 - y^2) \tag{44}$$

with *l* being the length of the pipe, *R* being the radius and *y* being a coordinate originating at the middle of the cross-section with *0 < y < R*. The definition of wall shear stress is:

$$\tau_w = \mu \cdot \left. \frac{\partial u}{\partial y} \right|_w \tag{45}$$

Pressure loss for laminar flow is defined in the following way:

$$\Delta p = \frac{\lambda \cdot l}{d} \cdot \frac{\rho}{2} \cdot u_\infty^2 = \frac{64}{Re} \cdot \frac{l}{d} \cdot \frac{\rho}{2} \cdot u_\infty^2 = \frac{32 \cdot l}{d} \cdot \frac{v \cdot \rho}{d} \cdot u_\infty \tag{46}$$

The derivative *δu/δy*, evaluated at the wall position *y = R*, can easily be calculated, and yields:

$$\left. \frac{\partial u}{\partial y} \right|_w = 2 \cdot R \frac{\Delta p}{4 \cdot \mu \cdot l} \tag{47}$$

Combining these results one obtains for the theoretical ratio of wall shear stress to the free-stream velocity:

$$\frac{\tau_w}{u_\infty} = 8 \cdot \frac{v \cdot \rho}{d} \tag{48}$$

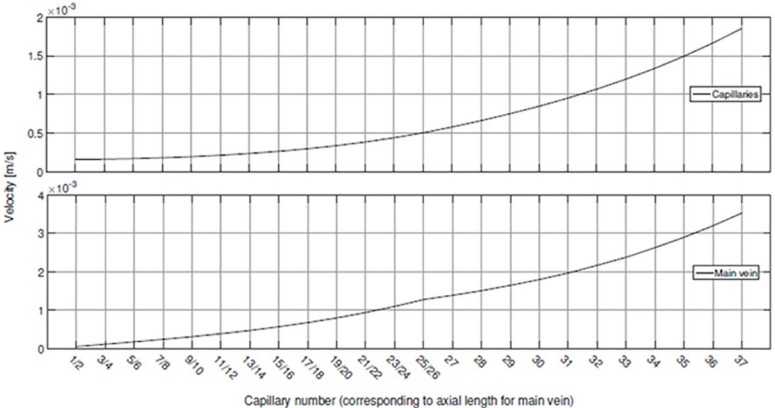

**Fig 13. Development of O₂ transport velocity in capillaries and main vein.**

This ratio is easily found by evaluating the wall shear stress for each capillary in Star-CCM+ resulting in values in the range of 46.5–47.2 Pa with a theoretical value of 47.7 Pa. The assumption of a quadratic wall shear stress along the main vein results in values in the range of 19.8–27.2 Pa with a theoretical value of 20.3 Pa, which is still a reasonable range. Furthermore, the error made by this assumption has very little influence on the results presented as follows.

## Distribution of oxygen mass fraction along capillaries and main vein

In the following section results are presented for the distribution of oxygen mass fraction along each capillary and along the main vein.

The development of velocity of the O₂ transport is shown for all capillaries at the top and along the main vein at the bottom of Fig 13. Capillaries are numbered in the direction from lung to heart starting with number 1 on the left side, number 2 on the right side (in frontal view), and so on. The opposing capillary pairs up to capillary number 26 (compare Fig 12) are displayed only once.

Velocity in the capillaries increases non-linearly (Fig 13, top), which is a consequence of pressure loss inside the vein system. Due to the pairwise inflow into the main vein up to capillary pair 25/26 velocity inside the main vein increases slower for axial positions corresponding to capillary numbers 27–37 (Fig 13, bottom) resulting in the change of slope that can be observed at capillary number 27. At the end of the main vein, corresponding to capillary number 37, the velocity $u_{main\ vein}$ = $0.0035\ m\cdot s^{-1}$ is reached according to the boundary condition.

Fig 14 shows the distribution of the solute mass fraction of oxygen after chemical bonding $\omega_2$ in the hemolymph of *X. derbentina* along the N = 40 segments of all capillaries. It can be observed that the progression of $\omega_2$ along each capillary proceeds in an asymptotical way and approaches the saturation value $\omega_{max}$. Values for $\omega_2$ are generally reduced for capillaries proximal to the heart. The reason for this effect is the aforementioned increasing velocity inside the downstream capillaries that results in the concurrent increase of mass flow into which the transferred diffusion mass flow of oxygen has to dissolve. Thereby, the mass fraction decreases. Another explanation for this observation is the lower contact time of the snail's hemolymph with the membrane surface that prevents a further rise in oxygen mass fraction.

However, this trend reveals an inflection point at the position of capillary 25/26 after which the slope decreases again and takes an asymptotical course towards the last capillaries (Fig 15). The reason for this can be found in the increase of mass fraction. The existence of an upper and lower limit for those mass fractions $\omega_2$ and $\omega_w$, namely $\omega_{max}$ and 0 or $\omega_{w1}$ for the first

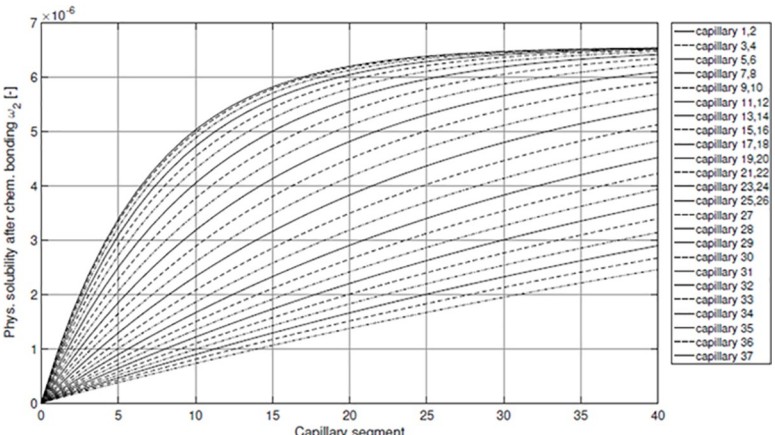

**Fig 14. Mass fraction of physical solute $O_2$ in the hemolymph along each capillary.**

segment, respectively, prohibits concentrations to exceed or fall below these values. However, these limits will not be reached suddenly but in an asymptotical way.

Related to the observations for the capillaries an increasing diffusion mass flow of oxygen results along the main vein (Fig 16). Due to the mass fraction decrease inside the main vein in downstream direction the gradient $\omega_{max-\omega w}$ increases, which results in a higher diffusion mass flow $J$ into the hemolymph towards the snail's heart. By the addition of the integrated mass flows for each capillary the diffusion mass flow for the complete vein system of *X. derbentina* is obtained.

Converting this value using the molar mass of oxygen $M_{O_2}$ a final value of

$$\dot{n}_{O_2} = 1.097 \cdot 10^{-10} mol \cdot s^{-1} \tag{49}$$

is obtained. Comparing this value with the measured oxygen consumption for medium-sized specimens at 25°C of $(1.04 +/- 0.25) \times 10^{-10}$ *mol· s⁻¹* shows an excellent agreement with the measurements. This suggests that the calculated value of the measured oxygen consumption

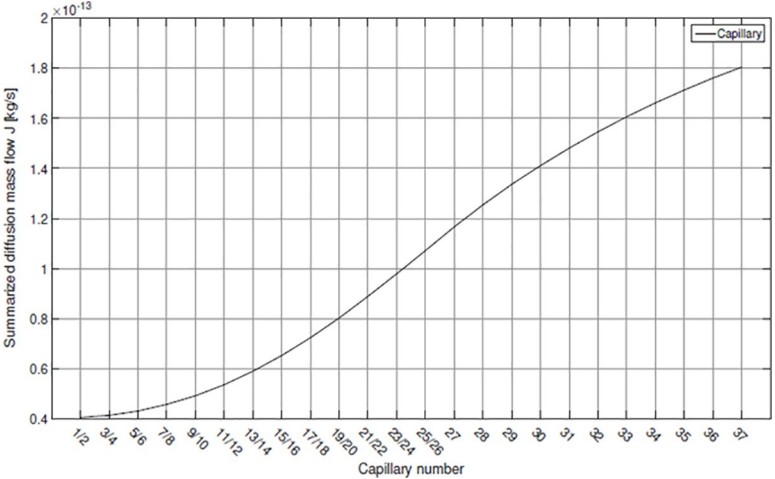

**Fig 15. Integrated diffusion mass flow $J$ for each capillary.**

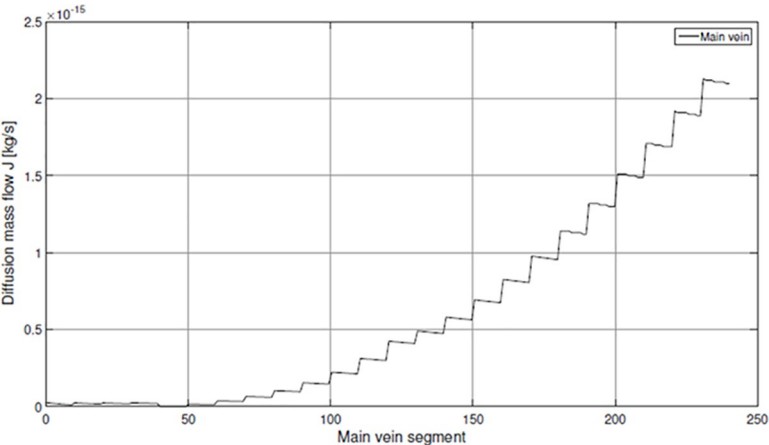

**Fig 16. Diffusion mass flow along the main vein.**

provides a meaningful proof for the correctness of the chosen values of the variables defined above and for the applied concept of the Colburn analogy as a whole.

## Limitations of the procedures and methods

The limitations of the presented methods become clear when examining the actual amount of oxygen dissolved in the hemolymph of *X. derbentina*. Based on the ventricle volume that has been segmented from the NMR measurements and the estimated heart rate for medium-sized specimens at 25 C ($f = 4.8 \ min^{-1}$) the volume flow of hemolymph was calculated to be approximately $2 \times 10^{-5} \ l \cdot min^{-1}$. This is also the exact volume flow that passes the lung of the animal after oxygen has been transported to the organs. The measured oxygen turnover ($1.04 \times 10^{-10} \ mol \cdot s^{-1} = 1.4 \times 10^{-7} \ l \cdot min^{-1}$), therefore, is the same amount that dissolves into this amount of hemolymph. The amount of dissolved ml of oxygen per 100 ml of hemolymph, therefore, results in:

$$c_{dissolved \ O_2} = \frac{1.4 \cdot 10^{-7} l \cdot min^{-1}}{2 \cdot 10^{-5} l \cdot min^{-1}} = 0.7 ml \cdot 100 ml^{-1} \tag{50}$$

According to Lide [40] the physical solubility of oxygen in water at 25˚C is 0.65 $ml \cdot 100 \ ml^{-1}$. Therefore, the rise of solute oxygen concentration due to chemical bonding onto hemocyanin results into 0.7/0.65 = 1.077, which is significantly lower than the defined value of $\Phi = 2.24$ used for the theoretical calculation of oxygen consumption based on the Colburn analogy. However, oxygen consumption was measured with an uncertainty of $0.25 \times 10^{-10} \ mol \cdot s^{-1}$.

Taking the highest value of $\dot{n}_{O_2} = 1.29 \times 10^{-10} \ mol \cdot s^{-1}$ into account the result rises to 1.34 and the discrepancy to $\Phi = 2.24$ decreases.

According to the procedure mentioned before a new heart rate range can be obtained in the range of 4–8 $min^{-1}$ corresponding to values of $\Phi = 4$ and $\Phi = 2$, respectively. This results in an increase in concentration due to chemical bonding of 1.9 for the lower heart rate and even a decrease of 0.95 for the higher frequency.

Both results show a distinct deviation from the theoretical values of $\Phi = 4$ and $\Phi = 2$. Taking the measured uncertainty into account again reduces this error but cannot fully eliminate it.

The reasons for these discrepancies are most likely caused by the calculation of the heart rate [44] that is based on the oxygen partial pressures according to the results of Mikkelsen and Weber [39] for *H. pomatia*. As these values are so far unknown for *X. derbentina* this

source of error cannot be eliminated. Furthermore, it is known that stroke volume in mollusks can change considerably according to metabolic fluctuations as the heart is not strictly rigid but its shape and volume depends rather on the internal pressures that provide structural stability [45, 61]. In addition, the heart adapts to the venous flow (preload) by varying its stroke volume [62, 63], which is also known as the Frank-Sterling mechanism. Smith and Hill [64] also reported a decrease of stroke volume by the addition of cardioactive drugs in *Busycon canaliculatum*. All these influences on cardiac output constitute a large source of error that is likely to be the reason for the previously discussed deviations. As the NMR measurements only provide a relatively undefined snapshot of the cardiac system of *X. derbentina* a slight change in heart volume is certainly relevant for the present investigation.

## Conclusions

A modelling approach is presented to validate the measured oxygen concentration by using an adjusted version of the Colburn analogy between wall shear stress inside the vein system and mass transfer of oxygen into *X. derbentina*'s hemolymph. This approach requires a much lower computational effort compared to a complex CFD simulation that directly takes the diffusive processes of oxygen into account. The suitability of the model became apparent by the agreement of simulated oxygen consumption minima (dependent on temperature and shell-free weight) with the observable limits of inactivity (either arousal or maximum temperature stress responses), even though some optimization may be needed to avoid negative $O_2$ consumption in the models. Each capillary and the main vein were divided into segments of finite lengths. For the first segment of each capillary as well as for the first segment of the main vein the oxygen mass fraction inside the hemolymph was chosen to be zero as a starting condition. Subsequently the oxygen mass fraction at the inner membrane wall of the first segment was calculated using the *Colburn analogy* in combination with the constant and known value of oxygen mass fraction inside the snail's lung. *Fick's law* was then utilized to obtain the absolute transferred diffusion mass flow of oxygen inside the segment. By passing this value to the next segment and relating it to the mass flow of hemolymph in this segment a new value for oxygen mass fraction inside the snail's hemolymph was calculated for the second element. This enabled the calculation of the oxygen mass fraction at the membrane wall for the second element and provides the necessary value to start the iteration process.

By successively passing the local oxygen diffusion mass flows to the next segment and relating the summarized oxygen diffusion mass flows to the mass flow of hemolymph in the following segment an incrementing solution algorithm was obtained.

This algorithm depends on the choice of appropriate starting and boundary conditions and was used to present a course for oxygen mass fractions as well as the transferred oxygen diffusion mass flows along each capillary and main vein segment. The evaluation of the absolute transferred oxygen mass flow obtained by the presented method reveals a very good agreement with the measured oxygen consumption of medium-sized specimens at 25˚C and is a good indicator for the correctness of the method.

The successful application of the Colburn analogy to the simulation of a metabolic process offers new ways of modelling in biology. The comparison between a sophisticated CFD simulation and an analytic model shows in the case of a laminar flow a good agreement. As many diffusive processes are characterized by laminar flow at low Reynolds numbers, a one-dimensional analytical modelling enables a straightforward description of the basic phenomena and processes. A validated simulation model enables the discussion of environmental influences on the metabolism such as temperature and the related physical hemolymph properties. Herewith, we provide a rather simple mathematical tool with which a simulation of allometric

growth will be possible that can address questions on the optimization of the respiratory system of land snails in general: Did evolution optimize the vein system in *X. derbentina* or could it be, theoretically, improved to withstand higher environmental temperatures? Should larger land snail species rather evolve either higher numbers or greater diameters of veins? Do limitations placed by anatomical features of the respiratory vein system set boundaries for the maximum size of snails in a particular climate? In summary, we consider our work a suitable example for the potential of mechanistically oriented modelling of morphologically confined physiological processes to understand ecological limitations of taxa that are trapped in a given body plan.

## Abbreviations

### Greek symbols

| | | |
|---|---|---|
| $\alpha$ | allometric scaling exponent | — |
| $\alpha$ | heat transfer coefficient | $W \cdot m^{-2 \cdot K^{-1}}$ |
| $\beta$ | mass transfer coefficient | $m \cdot s^{-1}$ |
| $\Delta H_{vap}$ | molar enthalpy of vaporization | $J \cdot mol^{-1}$ |
| $\Delta O_{2breath}$ | oxygen reduction per breath | $\%$ |
| $\lambda$ | thermal conductivity | $W \cdot m^2 \cdot K^{-1}$ |
| $\eta$ | dynamic viscosity | $Pa \cdot s$ |
| $v$ | kinematic viscosity | $m^2 \cdot s^{-1}$ |
| $\Phi$ | relative humidity | $\%$ |
| $\rho$ | density | $kg \cdot m^{-3}$ |
| $\omega$ | mass fraction | — |
| $\tau$ | shear stress | $N \cdot m^{-2}$ |
| $\Delta\omega$ | decrease of $O_2$ mass fraction in the lung | — |

### Roman symbols

| | | |
|---|---|---|
| $A$ | area | $m^2$ |
| $a$ | thermal diffusivity | $m^2 \cdot s^{-1}$ |
| $b$ | allometric coefficient | — |
| $c$ | concentration | $mol \cdot m^{-3}$ |
| $cf$ | friction coefficient | — |
| $c_v$ | specific heat capacity at constant volume | $J \cdot kg^{-1} \cdot K^{-1}$ |
| $c_p$ | specific heat capacity at constant pressure | $J \cdot kg^{-1} \cdot K^{-1}$ |
| $D$ | diffusion coefficient, diffusivity | $m^2 \cdot s^{-1}$ |
| $D$ | diameter | $m$ |
| $d$ | distance | $m$ |
| $e(T)$ | saturated vapor pressure | $Pa$ |

| $f$ | frequency | Hz |
|---|---|---|
| $f_b$ | breathing frequency | $\text{min}^{-1}$ |
| $J$ | diffusion mass flow | $\text{kg} \cdot \text{s}^{-1}$ |
| $j$ | diffusion flux | $\text{mol} \cdot \text{m}^{-2} \cdot \text{s}^{-1}$ |
| $k_{H,cp}$ | Henry constant | $\text{mol} \cdot \text{l}^{-1}\text{atm}^{-1}$ |
| $L$ | characteristic length | m |
| $l$ | length | m |
| $m$ | mass, body weight | kg |
| $MR$ | metabolic rate | $\text{mol} \cdot \text{s}^{-1}$ |
| $n_{O_2}$ | oxygen consumption | $\text{mol} \cdot \text{s}^{-1}$ |
| $n$ | amount of substance | mol |
| $n$ | number of specimens, number of segments | — |
| $p$ | pressure | Pa |
| $p$ | probability value | — |
| $R$ | radius | m |
| $R$ | universal gas constant | $\text{J} \cdot \text{mol}^{-1} \cdot \text{K}^{-1}$ |
| $r$ | correlation coefficient | — |
| $r$ | ratio of lung volume to breathing volume | — |
| $r$ | spatial fraction of substances | — |
| $R^2$ | coefficient of determination | — |
| $T$ | temperature | K |
| $t$ | time | s |
| $u$ | velocity in x direction | $\text{m} \cdot \text{s}^{-1}$ |
| $V$ | volume | $\text{m}^3$ |
| $V_b$ | breathing volume | $\text{m}^3$ |
| $V_l$ | lung volume | $\text{m}^3$ |
| $V_m$ | molar volume | $\text{m}^3 \cdot \text{mol}^{-1}$ |
| $X$ | Concentration fraction | — |

## Dimensionless numbers

| Pr | Prandtl number | $Pr = \dfrac{\eta \cdot c_p}{\lambda}$ |
|---|---|---|
| Re | Reynolds number | $Re = \dfrac{u \cdot L}{v}$ |
| Sc | Schmidt number | $Sc = \dfrac{v}{D}$ |
| Sh | Sherwood number | $Sh = \dfrac{\beta \cdot L}{D} = \dfrac{j_w \cdot L}{\rho \cdot D \cdot \Delta\omega}$ |

## Acknowledgments

We like to thank Andreas Dieterich, Günter Gauglitz, Markus Ludwig, Frank Bühler and Joachim Herz for collaboration and Bruker BioSpin GmbH for supporting the NMR measurements. Furthermore, we are very grateful to Yvan Capowiez and Rita Triebskorn for their help in providing *X. derbentina* from their natural habitat.

## Author Contributions

**Conceptualization:** Heinz-R. Köhler, Ulrich Gärtner.

**Data curation:** Ulf Fischbach.

**Investigation:** Ulf Fischbach.

**Supervision:** Heinz-R. Köhler, David Wharam, Ulrich Gärtner.

**Writing – original draft:** Ulf Fischbach.

**Writing – review & editing:** Heinz-R. Köhler, David Wharam, Ulrich Gärtner.

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
