## [Decision Letter · Decision Letter 0]

20 Apr 2021

PONE-D-21-05645

Modeling the oxygen uptake, transport and consumption in an estivating terrestrial snail, Xeropicta derbentina, by the Colburn analogy

PLOS ONE

Dear Dr. Köhler,

Thank you for submitting your manuscript to PLOS ONE. After careful consideration, we feel that it has merit but does not fully meet PLOS ONE’s publication criteria as it currently stands. Therefore, we invite you to submit a revised version of the manuscript that addresses the points raised during the review process.

We look forward to receiving your revised manuscript.

Kind regards,

David L. Hu

Academic Editor

PLOS ONE

Journal Requirements:

Reviewers' comments:

Reviewer's Responses to Questions

**Comments to the Author**

1. Is the manuscript technically sound, and do the data support the conclusions?

Reviewer #1: Yes

2. Has the statistical analysis been performed appropriately and rigorously? 

Reviewer #1: Yes

3. Have the authors made all data underlying the findings in their manuscript fully available?

Reviewer #1: Yes

4. Is the manuscript presented in an intelligible fashion and written in standard English?

Reviewer #1: Yes

5. Review Comments to the Author

Reviewer #1: This work provided a well-established framework combining the lumped analysis model (based on the momentum and mass transport in a wall-bounded shear flow) and numerical simulation, aiming at investigating the complex metabolic system. I have enjoyed reading the paper and the approaches were explained in detail and the results were discussed clearly. I can accept this for the publication with a minor comment. In equation (32), the diffusion coefficient D_1 was adopted for that in the water, as the authors specified. Can it be further justified?

6. PLOS authors have the option to publish the peer review history of their article (what does this mean?). If published, this will include your full peer review and any attached files.

Reviewer #1: No

---

## [Author Response · Author response to Decision Letter 0]

21 Apr 2021

Response to Reviewers

Manuscript:

Modeling the oxygen uptake, transport and consumption in an estivating terrestrial snail, Xeropicta derbentina, by the Colburn analogy

by

Ulf Fischbach, Heinz-R. Köhler, David Wharam, Ulrich Gärtner

Dear Editors, 

Please find our revised ms. Apart from praise, there was just a single suggestion for improvement:

Reviewer 1: „I can accept this for the publication with a minor comment. In equation (32), the diffusion coefficient D_1 was adopted for that in the water, as the authors specified. Can it be further justified?“ 

We gladly take up this suggestion and added an explanatory sentence in the revised MS (page 19, lines 2-4 in the „Track Changes“ version; page 18, last line and page 19, lines 1-2 in the „Changes Accepted“ version): 

„This approximation is justified by the fact that, besides of some cells and dissolved ions, carbohydrates, lipids, glycerol, amino acids, hormones and pigments, hemolymph is basically water.“

Furthermore, we formatted our ms according tot he formatting guidelines. We hope that our revised manuscript can now be accepted for publication.

With best regards, 

Sincerely,

Heinz Köhler

---

## [Decision Letter · Decision Letter 1]

26 Apr 2021

Modeling the oxygen uptake, transport and consumption in an estivating terrestrial snail, Xeropicta derbentina, by the Colburn analogy

PONE-D-21-05645R1

Dear Dr. Kohler,

We’re pleased to inform you that your manuscript has been judged scientifically suitable for publication and will be formally accepted for publication once it meets all outstanding technical requirements.

Kind regards,

David L. Hu

Academic Editor

PLOS ONE

Additional Editor Comments (optional):

Reviewers' comments:

Reviewer's Responses to Questions

**Comments to the Author**

1. If the authors have adequately addressed your comments raised in a previous round of review and you feel that this manuscript is now acceptable for publication, you may indicate that here to bypass the “Comments to the Author” section, enter your conflict of interest statement in the “Confidential to Editor” section, and submit your "Accept" recommendation.

Reviewer #1: All comments have been addressed

2. Is the manuscript technically sound, and do the data support the conclusions?

Reviewer #1: Yes

3. Has the statistical analysis been performed appropriately and rigorously? 

Reviewer #1: Yes

4. Have the authors made all data underlying the findings in their manuscript fully available?

Reviewer #1: Yes

5. Is the manuscript presented in an intelligible fashion and written in standard English?

Reviewer #1: Yes

6. Review Comments to the Author

Reviewer #1: Authors have explained the issue that I have raised for the previous version. Now, I can recommend this for the publication.

7. PLOS authors have the option to publish the peer review history of their article (what does this mean?). If published, this will include your full peer review and any attached files.

Reviewer #1: No

---

## [Editor Report · Acceptance letter]

28 Apr 2021

PONE-D-21-05645R1 

Modeling the oxygen uptake, transport and consumption in an estivating terrestrial snail, *Xeropicta derbentina*, by the Colburn analogy 

Dear Dr. Köhler:

I'm pleased to inform you that your manuscript has been deemed suitable for publication in PLOS ONE. Congratulations! Your manuscript is now with our production department. 

Kind regards, 

on behalf of

Dr. David L. Hu 

Academic Editor

PLOS ONE